

# Characterization of Shallow Oceanic Precipitation using Profiling
and Scanning Radar Observations at the Eastern North Atlantic ARM
Observatory
Katia Lamer[1], Bernat Puigdomènech Treserras[2], Zeen Zhu[3], Bradley Isom[4], Nitin Bharadwaj[4], and
Pavlos Kollias[3,5]
[1.] Department of Earth and Atmospheric Science, The City College of New York
[2.] Department of Atmospheric and Oceanic Sciences, McGill University
[3.] School of Marine and Atmospheric Sciences, Stony Brook University
[4.] Atmospheric Measurement and Data Sciences, Pacific Northwest National Laboratory
[5.] Department of Environmental and Climate Sciences, Brookhaven National Laboratory
Correspondance: Katia Lamer, klamer@ccny.cuny.edu
**Abstract**

18        Shallow oceanic precipitation variability is documented using 2nd generation radars located
at the Atmospheric Radiation Measurement (ARM) Eastern North Atlantic observatory: the Ka-
band ARM zenith radar (KAZR2), the Ka-band scanning ARM cloud radar (KaSACR2) and the
X-band scanning ARM precipitation radar (XSAPR2). First, the radars and measurement post-
processing techniques, including sea clutter removal and calibration against collocated
disdrometer and Global Precipitation Mission (GPM) observations are described. Then, we present
how a combination of profiling radar and lidar observations can be used to estimate adaptive (in
both time and height) parameters that relate radar reflectivity ($Z$) to precipitation rate ($R$) in the
form $Z = \alpha R^{\beta}$ which we use to estimate precipitation rate over the domain observed by XSAPR2.
Furthermore, Constant Altitude Plan Position Indicator (CAPPI) gridded XSAPR2 precipitation
rate maps are also constructed.
Hourly precipitation rate statistics estimated from the three radars differ; that is because KAZR2
is more sensitive to shallow virga and because XSAPR2 suffers from less attenuation that
KaSACR2 and as such is best suited to characterize intermittent and mesoscale-organized
precipitation. Further analysis reveals that precipitation rate statistics obtained by averaging 12h
of KAZR2 observations can be used to approximate that of a domain of 2,500 km$^2$ averaged over
similar time periods. However, it was determined that KAZR2 is unsuitable to characterize domain
average precipitation rate over shorter periods. But even more fundamentally, these results suggest
that observations cannot produce objective domain precipitation estimate and that forward-
simulators should be used to guide high temporal-resolution model evaluation studies.



## 1.0 Introduction

Characterizing shallow oceanic precipitation is all-important to improving our understanding of shallow cloud systems since precipitation is related to a number of cloud process all of which may affect cloud properties. For example, precipitation leads to a reduction in the droplet number via the collision-coalescence process and of the liquid water path through sedimentation. Furthermore, a number of modeling studies have suggested that drizzle organization, intensity and subcloud layer evaporation could play a role in organizing stratocumulus cloud decks on the mesoscale (Zhou et al., 2017; Savic-Jovcic and Stevens, 2008; Wang and Feingold, 2009; Yamaguchi and Feingold, 2015; Zhou et al., 2018). Ultimately, these controls may alter low cloud radiative properties and climate (Wood, 2012). Quantification, over a domain of several kilometers, of marine drizzle cell precipitation rate, along with sub cloud layer evaporation rate, thermodynamic properties and dynamics could provide additional observational constrains for modeling studies unfortunately, such observations remain challenging to collect over the ocean.

Although satellite-based microwave sensors can infer the spatial distribution of liquid water path (Wood and Hartmann, 2006; Miller and Yuter, 2013) and precipitation rate (Ellis et al., 2009; Adler et al., 2009; Rapp et al., 2013) they have poor horizontal resolution and suffer from surface inference causing them to under sample the cloud field variability and to underreport boundary-layer cloud and precipitation occurrence (Schumacher and Houze Jr, 2000; Rapp et al., 2013). In contrast, airborne (Stevens et al., 2005; Wood et al., 2011; Moyer and Young, 1994; Vali et al., 1998; Paluch and Lenschow, 1991; Sharon et al., 2006) and ship-based (Yuter et al., 2000; Comstock et al., 2005; Feingold et al., 2010) sensors can resolve the spatial/temporal variability of the cloud and precipitation field, but field campaigns deploying such sensors are often expensive to conduct and limited in temporal duration (Stevens et al., 2003; Bretherton et al., 2004; Rauber et al., 2007). Island-based observatories such as the U.S. Department of Energy (DOE) Atmospheric Radiation Measurement (ARM) Eastern North Atlantic observatory (ENA, Mather et al., 2016; Kollias et al., 2016) and the Barbados Cloud Observatory (BCO, Lamer et al., 2015; Stevens et al., 2016) operating profiling and scanning remote sensors can provide long-term statistics of marine light precipitation.

Beyond detecting, quantifying precipitation rate from warm clouds is especially challenging since the droplets they contain do not generate the typical polarimetric signals required of common precipitation rate retrievals (e.g., Villarini and Krajewski, 2010; Gorgucci et al., 2000). As an alternative to polarimetric signatures, a combination of sensors is typically required to retrieve precipitation rate ($R$); Combinations of radar reflectivity ($Z$) and in-situ measurements have led to the development of $Z$-$R$ relationships (Wood, 2005; Comstock et al., 2004; VanZanten et al., 2005; Vali et al., 1998) however, these tend not to be universally applicable since they are based on assumptions about the drizzle particle size distribution which may vary with factors such as aerosol loading and liquid water path. Moreover, relying on surface disdrometer measurements to characterize warm precipitation may be especially unsuitable at the ENA where i) a large fraction of the precipitation does not reach the surface (Yang et al., 2018), ii) precipitation reaching the ground typically does so with an intensity below the detection limit of most optical-based disdrometers ($\sim 10^{-2}$ mm hr$^{-1}$) and iii) evaporation is an active process such that water drop size distribution information retrieved at one height may not be appropriate to represent the entire





atmospheric column. Alternatively, a method combining radar reflectivity and lidar backscatter
measurements has been proposed to retrieve $R$ without assumptions about the drizzle particle size
distribution (Intrieri et al., 1993; O'Connor et al., 2005); Because of the rarity of scanning lidar
observations, this technique has only been used to retrieved $R$ in the column and cannot be used to
address the concerns present in recent studies suggesting that scanning systems are essential to
map domain properties (Oue et al., 2016).
Here we propose to exploit the availability of collocated vertically-pointing radar and lidar as well
as scanning radar systems to characterize marine precipitation rate variability over a domain of 40-
60 km around the ENA observatory. The ENA, with its abundance of marine boundary layer
precipitating clouds, is an ideal location for such study (Rémillard and Tselioudis, 2015; Wood,
2012).Observations from the Ka-band ARM Zenith Radar (KAZR2) and zenith-pointing
ceilometer lidar are combined to estimate adaptive (both in time and height) *Z-R* relationships
which we then use to estimate precipitation rate across the domain observed by the X-band
Scanning ARM Precipitation Radar (XSAPR2). Domain-average and time-average precipitation
rate estimates obtained from zenith-pointing and scanning observations are compared to document
the complementarity and applicability of each sensor in documenting precipitation rate from warm
boundary layer clouds.
**2.0 Eastern North Atlantic Observatory**
In October 2013, the ARM program established a permanent observatory in the Eastern North
Atlantic on the island of Graciosa (~60 km$^2$ area; 39.1°N, 28.0°W). The site, located within the
Azores archipelago, straddles the boundary between the subtropics and the midlatitudes and as
such is subject to a wide range of different meteorological conditions including periods of
relatively undisturbed trade-wind flow, midlatitude cyclonic systems and associated fronts, and
periods of extensive low-level cloudiness (Rémillard and Tselioudis, 2015). The observatory hosts
an extensive instrument suite including three second generation radar systems: the Ka-band ARM
Zenith Radar (KAZR2), the dual-frequency Ka-and W-band Scanning ARM Cloud Radar
(SACR2) and the X-band Scanning ARM Precipitation Radar (XSAPR2) which's specifications
are listed in Table 1. A short description of the radar systems is provided here with emphasis on
changes in configuration from the first to the second generation.
**2.1 KAZR2**
KAZR2 operates at 34.8 GHz ($\lambda$ = 8.6 mm) and is an upgraded version of the KAZR that
replaced the ARM MilliMeter Cloud Radar (MMCR, Kollias et al., 2016). KAZR2 uses an
Extended Interaction Klystron (EIK) amplifier with a 2.2 kW peak power and 5 % duty cycle. Its
dual receiver configuration allows the simultaneous transmission of a long (4 µs) pulse with
frequency modulation (pulse compression) for higher sensitivity (~-44 dBZ at 1 km not
considering signal integration gain) at ranges from 737 m from the radar to 18 km and a short pulse
(200 ns) with a sensitivity of (~-32.5 dBZ at 1 km not considering signal integration gain) at ranges
from 72 m to 18 km. KAZR2 has a narrow (0.3°) 3-dB antenna bandwidth and is nominally
operated with a range resolution of 30 m, a temporal resolution of 2 sec and is set to record the full
radar Doppler spectrum with 256 or 512 FFT points. KAZR2 transmits a horizontal pulse and
receives both horizontal and vertical polarization such that the only polarimetric information it can



measure is linear depolarization ratio.
**2.2 KaSACR2**
KaSACR2 is a fully polarimetric radar that operates at 35.3 GHz (λ = 8.5 mm) and is an
upgraded version of the single polarization KaSACR described in Kollias et al., (2014a,b). The
KaSACR2 also uses an EIK amplifier with a 2.2 kW peak power, has a 5 % duty cycle and a 3-dB
antenna beamwidth of 0.3°. Currently, it is operated with a short pulse, although it could be
operated with a longer pulse with pulse compression for increased sensitivity. Owing to its narrow
beam width KaSACR2 must scan rather slowly (3-6° s$^{-1}$) to collect observation with a sensitivity
of ~-15 dBZ at 20 km (not considering signal integration gain). The KaSACR2 conducts a cloud
sampling strategy that includes different modes (Kollias et al., 2014a,b). Here, because of our
interest to map precipitation structure and rate over a large horizontal domain, we only use
observations collected in Plan Position Indicator (PPI) configuration at 0.5° elevation angle over
a 160° wide azimuth sector. The KaSACR2 conducts a PPI scan every 15 min and takes 2 min to
collect each PPI. The KaSACR2 employs frequency hopping and staggered pulse repetition time
techniques to mitigate artifacts due to second trip echoes and velocity aliasing; This however
comes at the expense of preventing the collection of the full Doppler spectrum.
**2.3 XSAPR2**
XSAPR2 operates at 9.5 GHz (λ = 3.2 cm); It is an upgraded version of the XSAPR as it
operates with an improved digital receiver and a larger antenna (5 m) which results to an
exceptionally narrow 3-dB antenna beamwidth of 0.45°. The requirement for the XSAPR2 to have
a narrow antenna beamwidth emerged from a need to reduce the impact of sea-clutter at low-
elevations and maintain high angular resolution over a 60 km radius in order to resolve small scale
oceanic precipitating clouds. XSAPR2 uses a high-power Magnetron with a 300kW peak power
and a maximum duty cycle of 0.1 %. Under nominal operational conditions, the XSAPR2 transmits
a 60 m long pulse and scans at a relatively slow rate (6° s$^{-1}$) to collect observations with a sensitivity
of ~-21 dBZ at 20km (not considering integration gain). The XSAPR2 volume coverage pattern
(VCP) scan strategy consists of a series of PPI scans every 0.5° elevation between the angles of 0°
and 5°. Because of considerable beam blockage in the southerly direction a 160° azimuth sector
coverage is achieved. The VCP scan (i.e. the entire set of PPI scans) is completed within 5 min
and subsequently repeated. Horizontal and vertical polarization are possible for both transmit and
receive states, meaning XSAPR2 collects a full suite of polarimetric variables while in scanning
mode.
**3.0 Radar Observations Post-Processing**
Radar observations require considerable post-processing for the removal of non-
meteorological targets before they can be scientifically interpreted or used to retrieve geophysical
quantities such as precipitation rate. Radar data post-processing is described in section 3.1 and
cross-comparison between different systems for calibration is described in section 3.2. Note that
the KAZR2 data used for analysis are from "enakazrgeC1.a1" files, KaSACR2 data are from
"enakasacrppivhC1.a1" files and the XSAPR2 from the "enaxsaprsecD1.00 files". All data files
were obtained from the ARM archive (https://www.archive.arm.gov/discovery/).



## 3.1 Removal of Non-Meteorological Targets

First, signal processing artifacts (e.g. second trip echoes) and echoes of non-meteorological origin (e.g., biological echoes, sea-clutter, and ground-clutter) are identified and removed.

The KaSACR2 system operates in fully polarimetric mode and uses staggered pulse repetition time and frequency hopping to automatically remove second trip echoes, perform velocity dealiasing and increase the number of independent samples (Pazmany et al., 2013). The XSAPR2 systems operates using a magnetron system which is coherent on receive (i.e., transmitted pulse phase is random). For the XSAPR2, the removal of second trip echoes is done using Normalized Coherent Power (NCP) which is the coherency of the received pulse with respect to the last transmitted pulse. For atmospheric echoes within maximum unambiguous range, NCP is high since the radar receiver is phase-locked to the phased of the last transmitted pulse. Outside of the maximum unambiguous range, NCP is low since the radar receiver has already phase-locked on the phase of another transmitted pulse. Here, an NCP threshold of 0.3 is used to identify echoes originating from outside the maximum unambiguous range (i.e. second trip echoes).

Biological targets such as insect and birds often contaminate radar observations especially over land (e.g., Luke et al., 2008). Their occurrence varies with atmospheric condition, time of the year, and time of the day (Alku et al., 2015). KAZR2 observations at the ENA seem minimally impacted by biological echoes. Furthermore, the fact that the bulk of the KaSACR2 and XSAPR2 observations are collected over open ocean and that Graciosa is a small island suggests that biological targets should not be a concern at this particular location.

On the other hand, low elevation angle observations are susceptible to sea-clutter contamination. Research on radar sea-clutter characterization and remediation has been ongoing for over 20 years (e.g., Horst et al., 1978; Gregers-Hansen and Mital, 2009; Nathanson et al., 1991); Observational and modeling studies suggest that factors such as oceanic wave properties (related to local wind speed and direction), swell and air density streams can affect sea-clutter occurrence. Radar characteristics such as wavelength, wave polarization, beam width and grazing angle are also known to affect sea-clutter characteristics, amounts and our ability to isolate atmospheric returns from sea-clutter. Here, observations collected over a range of wind conditions during nearly 100 hours of clear sky conditions are used to examine how sea-clutter characteristics vary with radar wavelength, beam width and beam elevation angle.

First, the distribution of sea-clutter reflectivities as measured by the XSAPR2 and KaSACR2 at elevation 0.5° are compared to document the antenna beam width effect (Fig. 1d). The KaSACR2 (0.3° 3-dB antenna beam width) sea-clutter reflectivity distribution is narrower with a peak at -21 dBZ and a majority of echoes below -15 dBZ (Fig. 1d black line) while the XSAPR2 (0.45° 3-dB antenna beam width) sea-clutter reflectivity distribution is wider, peaks at -18 dBZ and covers a range from -40 dBZ to +10 dBZ (Fig. 1d red line). This can be explained by the XSAPR2 wider antenna beam width which results in a larger fraction of the radiated energy to hit ocean waves, causing higher ocean clutter return power. Similar to beam width, elevation angle affects how much sea is in the radar field of view and the spatial extent of observed sea-clutter. Figure 1d, shows that, at 1.0° elevation, XSAPR2 sea-clutter reflectivity peaks at a lower reflectivity of -25 dBZ (blue line) and Fig. 1b$_3$ shows that in this configuration it frequently (> 25 % of the time)





detects clutter only over a domain of 10 km radius around the site which is much less than it detects
when collecting observations at 0.5° elevation  (significant clutter in a 20 km radius around the
site Fig. 1a$_3$).
Now that we have characterized sea-clutter intensity and frequency of occurrence using clear sky
observations we next evaluate its impact on the detection of meteorological targets using
observations containing mixture of hydrometeor and sea-clutter. To isolate hydrometeors from
clutter, we exploit the correlation coefficient $\rho_{HV}$ which we know is affected by the relative
occurrence of signal to clutter; $\rho_{HV}$ is typically close to 1 for liquid-phase hydrometeors and lower
for non-meteorological targets. Looking at KaSACR2 reflectivity and $\rho_{HV}$ confirms that at Ka-
band wavelength the signal to clutter ratio is high and hydrometeors contributions dominate both
radar reflectivity and correlation coefficient measurements (Fig. 1c$_1$ and 1c$_2$, respectively). The
enhanced KaSACR2 signal-to-clutter ratio is attributed to two effects: i) its narrow beamwidth
which causes a smaller fraction of the transmitter energy to hit the sea surface and ii) its shorter
wavelength which creates a larger distinction between hydrometeor scattering - which follow
Rayleigh scattering $\sim 1/\lambda^4$ - and sea-clutter scattering – which follow $\sim 1/\lambda$ -.Using KaSACR2
observations has a guide to locate cloud and precipitation location (Fig 1c$_1$), it is apparent that it
is not possible to distinguish atmospheric signals from sea-clutter in XSAPR2 radar reflectivity
observation collected at 0.5° (Fig 1a$_1$).
Several techniques that use both time-domain and frequency domain filtering methods have been
proposed to discriminate between sea-clutter and meteorological targets in precipitation radar
observations (e.g., Torres and Zrnic, 1999; Siggia and Passarelli, 2004; Nguyen et al., 2008; Alku
et al., 2015). Ryzhkov et al. (2002) present an echo classification technique based on fuzzy logic
and a multiparameter dataset including radar reflectivity, mean Doppler velocity, spectrum width,
differential reflectivity, differential phase, linear depolarization ratio, and cross-correlation ($\rho_{HV}$).
In the current study, given the radars narrow beam width and short wavelength, an approach solely
based on $\rho_{HV}$ is used to filter sea-clutter. Since cross-correlation between horizontal and vertical
cross-polar received powers is largest for spherical hydrometeors, we label observations with
$\rho_{HV}$ larger than a certain threshold as atmospheric returns and the rest as sea-clutter. The analysis
of a large sample of $\rho_{HV}$ observations during clear and cloudy sky conditions indicates that the use
of a threshold of 0.9 for KaSACR2 and an average (over 5 range gates and 5 azimuthal
measurements) threshold of 0.55 for the XSAPR2 can be used to isolate hydrometeor-dominated
from clutter-dominates observations. The proposed $\rho_{HV}$ technique successfully isolates
atmospheric returns at the same location for both the X-band at 1.0° elevation and the reference
Ka-band 0.5° elevation (Fig. 1b$_2$ and c$_2$ respectively; pink regions). However, it only identifies a
fraction of the atmospheric returns in the X-band 0.5° elevation observations. There, additional
filtering, beyond the scope of this study, would be required to suppress the remaining sea-clutter
and recover the missing atmospheric returns (see (Moisseev and Chandrasekar, 2009; Unal, 2009)
who propose advanced technique). Given this, XSAPR2 cross validation and precipitation rate
maps will be estimated using observations collected at 1.0° elevation since it offers the best
compromise between proximity to the surface and minimum sea-clutter contamination.
**3.2 Radar Calibration**
Calibrated reflectivity observations are necessary to perform quantitative precipitation rate



retrievals. Following Kollias et al. (2019), KAZR2 calibration is performed using collocated
surface-based Parsivel laser disdrometer equivalent radar reflectivity estimates during light
precipitation events as well as CloudSat observations collected over a small radius around the site.
We estimate that, during the period of interest (01/10/2018 to 04/01/2018), KAZR2 radar
reflectivity measurements are off by about +3-dB which we proceeded to correct for. The detailed
time-series of KAZR2 calibration offset is presented in Fig. 2a.
Comparison of total (Fig. 3a) and range resolved (Fig. 3b) histograms of radar reflectivity
measured by KAZR2 (pre-calibration) and KaSACR2 at zenith confirm that during the analysis
period the KaSACR2 matched KAZR2. For this reason, KaSACR2 radar reflectivity
measurements were also adjusted by the calibration constant depicted in Fig. 2a. Note how this
comparison between the KAZR2 and KaSACR2 was performed between 1.5 to 5 km to avoid any
differences in the reported radar reflectivities due to differences in how they detect ground/sea-
clutter.
Calibrating the XSAPR2 radar reflectivity measurements is more challenging since it does not
perform profiling observations and as such it cannot be benchmarked against disdrometer and
KAZR2 observations. Here, we assess the calibration of the XSAPR2 radar using observations
from the Global Precipitation Measurement (GPM) Ku-band frequency of the Dual-frequency
Precipitation Radar (DPR) when the satellite track crosses within a 245 km radius of the XSAPR2
radar site. For comparison, ground-based XSAPR2 reflectivity measurements are smoothed and
interpolated to the satellite sampling volume: The azimuth-range measurements are smoothed
using the 0.71° 3-dB beamwidth antenna weighting function of the GPM DPR (5-km footprint).
Nearest neighbor is then used to match the satellite measurements in the horizontal plane while
linear interpolation is used to match them in the vertical plane (Warren et al., 2018). Matched
XSAP2 radar reflectivity measurements are compared to GPM DPR corrected reflectivity
measurements (GPM product version V06A (Iguchi et al., 2010)). Considering differences in radar
sensitivity, radar reflectivity measurements with returns smaller than 14 dBZ are not considered
during the calibration procedure (Toyoshima et al., 2015) and only periods when both radars
coincidently detect significant precipitation are used to perform calibration. For the analysis
period, a total of 3 GPM overpasses with significant precipitation were observed for a total number
of 1516 data points for the comparison.
An example of concurrent XSAPR2 and GPM DPR radar reflectivity observations are shown in
Fig. 4a and c respectively. The example shows that both radars detected several shallow
precipitation cells with cloud top heights between 3 and 4 km (Fig. 3b). Beyond agreeing in their
location, both radars (XSAPR2 and GPM DPR) are found to agree on the reflectivity intensity of
these precipitation echoes. To confirm their agreement, we estimated Contour of Frequency by
Altitude Diagram (CFAD) of the differences in radar reflectivities between the matched XSAPR2
and GPM DPR for all 1516 available observations (Fig. 4b). This comparison shows no noticeable
difference (i.e., no bias) between 1.5 and 3.5 km. Though in the lowest kilometer GPM's DPR
tends to overestimate the near surface radar reflectivity (Fig. 4b), a scatter plot between the
matched GPM DPR and XSAPR2 radar reflectivities confirms the overall lack of significant bias
between the two radars (Fig. 4d). This leads us to conclude that, for the observation period between
01/10/2018 to 04/01/2018, the XSAPR2 was well calibrated and does not require any radar
reflectivity adjustments.



**4.0 Radar Reflectivity-Based Precipitation Rate Retrievals**

Distinct considerations must be taken to quantitatively retrieve precipitation rate from KAZR2, XSAPR2 and KaSACR2 measurements.

**4.1 KAZR2**

Intrieri et al. (1993) and later O'Connor et al. (2005) proposed a technique to constrain water drop size distribution using lidar backscatter (related to water drop cross-section) and radar Doppler spectral width (related to the width of the water drop size distribution). This radar-lidar technique can be used to estimate precipitation rate at all levels in the subcloud layer when collocated radar and ceilometer observations are available. We apply this technique to the vertically pointing ceilometer lidar and KAZR2 pair operating at the ENA. The O'Connor et al. (2005) technique requires ceilometer backscatter to be calibrated and remapped to the radar spatio-temporal resolution (here 2 s x 30 m). Ceilometer backscatter is calibrated following a variation of the O'Connor et al. (2004) technique by scaling observed path-integrated backscatter in thick stratocumulus to match theoretical cloud lidar ratio values. Satisfactory conditions for ceilometer backscatter calibration are identified as the first (in time) 20-min periods each day with standard deviation of lidar ratio smaller than 1.5. The observed backscatter during the "satisfactory 20-min period" are input to Hogan (2006)'s multi scattered model to determine a daily backscatter calibration factor. For days where satisfactory conditions are not observed, a climatological calibration factor of 1.35 is used to calibrate the observed backscatter. For the current analysis period, the ceilometer backscatter calibration constant was estimated to vary around 1.35+/- 0.08. (Fig 2b). Calibrated ceilometer backscatter is subsequently mapped on the KAZR2 time-height grid using a nearest neighbor approach.

This radar-lidar technique generates time-height maps of precipitation rate from 200 m above ground level to 90 m below cloud base height which are filtered for aerosol contamination. We use the clear-sky – according to KAZR - calibrated lidar backscatter signals as a reference for aerosol behavior, lidar calibrated backscatter values below the mean clear-sky calibrated backscatter value at each height, depicted as the black vertical line in Fig. 2c, are systematically removed from the analysis to leave only drizzle signals. In additional to aerosol contaminated returns, unphysical values with median diameter smaller than 10 $\mu$ m or equal or large to 1000 $\mu$ m are also removed from our analysis.

Two one-hour examples of cloud location (black dots) and precipitation rate estimated using this technique are shown in Fig. 5a and b. Because of evaporation, the most intense precipitation rates are observed near cloud base height and a significant fraction of the precipitation does not reach the surface and falls as virga.

**4.2 XSAPR2**

As previously mentioned, the estimation of the precipitation rate for the XSAPR2 i) cannot depend on the use of polarimetric observations, because of the absence of polarimetric signature from spherical drizzle drops and ii) cannot depend on the use of disdrometer-based estimates of the relationship between the radar reflectivity ($Z$) and the precipitation rate ($R$), because





observations collected at the surface may not be representative of other levels in the subcloud layer
especially at the ENA where evaporation is an active process.
To accommodate changes in drizzle drop size distribution with height which could be associate
for example to changes in aerosol loading or evaporation, we propose to construct adaptive (both
with time and height) $Z$-$R$ relationships in the form $Z = \alpha R^{\beta}$ from precipitation rates retrieved
through the KAZR-ceilometer technique (see section 4.1). Every 30 min, independently for every
level in the subcloud layer, retrieved zenith precipitation rates ($R$ in mm hr$^{-1}$) and calibrated KAZR
reflectivity ($Z$ in mm$^6$ m$^{-3}$) reported during a 12-h window around that time are related through the
relationship:
$$\log_{10}(Z) = \log_{10}(\alpha) + \beta \cdot \log_{10}(R) \qquad\qquad (1)$$
The prefactor $\alpha$ and exponent $\beta$ are estimated using a total least square regression technique only
considering $R$ between $10^{-3.5}$ and $10^{0.5}$ mm hr$^{-1}$ and only if at least 350 precipitation detections are
available. When too few observations are available, average (for the period of the current study)
$\alpha$ and $\beta$ are used. A 12h time window was determined to be the best compromise between data
density and least change in water drop size distribution characteristics.
To evaluate the adaptive $Z$-$R$, we apply three different precipitation retrieval techniques to
KAZR2 reflectivity observations: We compare precipitation rate statistics retrieved following
the O'Connor et al. (2005) technique (ideal technique, red), to those estimated using $Z$-$R$
relationships constructed using fixed (approach proposed by Comstock et al. (2004), green) or
adaptive (approach proposed here, black) coefficients (presented in Fig. 6e and f respective).
Figure 6f shows that the proposed adaptive $Z$-$R$ relationships can reproduce the precipitation
rate statistics obtained using the ideal O'Connor et al. (2005) technique. The same cannot be
said from using traditional fixed $Z$-$R$ relationships such as that proposed by Comstock et al.
(2004) which tends to create an underestimation of precipitation intensity (Fig. 6e).
Fig. 6a and b respectively present time series of $\alpha$ and $\beta$ near cloud base (i.e., 90 m below
cloud base height) for a 30-day long period that overlaps with the second phase of the ACE-
ENA field campaign: Again for comparison we illustrate our adaptive coefficients (black), the
Comstock et al. (2004) constant coefficients (dashed green) and coefficients estimated from
surface-based Parsivel laser disdrometer measurements (dashed orange). The gradual increase
in both the adaptive $\alpha$ and $\beta$ coefficients over time is consistent with reports of observed
conditions indicating a transition from shallow precipitation at the end of January to deep
frontal precipitation at the end of February. CFADs of $\alpha$ and $\beta$ (Fig. 6c and d respectively)
show how the adaptive $\alpha$ additionally has a tendency to increase with distance from cloud base
(from top to bottom), which is consistent with the evaporation of small drops that leads to an
increase in mean drop size and has been previously reported by Comstock et al. (2004) and
discussed in VanZanten et al. (2005).
Figure 5c and d show how, by applying the adaptive $Z$-$R$, XSAPR2 reflectivity observations
collected at 1° elevation can be converted to precipitation rate. Note how the adaptive $Z$-$R$
relationships can be directly applied to clutter-filtered calibrated XSAPR2 radar reflectivity
measurements since two-way gas attenuation at X-band is negligible (generally amounts to 0.03


dB km$^{-1}$ according to Rosenkranz (1998)).
**4.3 KaSACR2**
Before quantitatively estimating precipitation rate from KaSACR radar reflectivity
measurements, we also consider how its wavelength responds to the presence of atmospheric
gases. Rosenkranz (1998) propagation model suggests that, for the conditions observed at the
ENA, two-way gas attenuation of Ka-band signals can amount to 0.25 dB km$^{-1}$. Although this may
seem small and can be insignificant when collecting observations of boundary layer clouds in
profiling mode, in scan mode, attenuation of Ka-band reflectivity by atmospheric gas can amount
to 10 dB at 40 km range (Fig. 9 difference between the black and green curve) and as such
should not be neglected. Also note that in addition to the gaseous attenuation, Ka-band radars
suffer from considerable liquid water attenuation. According to Matrosov (2005), the
relationship between one way liquid attenuation $a$ (dB km$^{-1}$) and precipitation rate $R$ (mm hr$^{-1}$)
is very robust ($a = 0.28R$). His findings were verified using Mie scattering calculations on
all particle size distributions observed by the ENA Parsivel laser disdrometer. Fig. 7e
illustrates an example of observations collected by the KaSACR at 0.5 elevation on
02/13/2018, in this example, liquid contributed anywhere from 2 to 10 dB in total attenuation
at Ka-band over the 40 km observation domain. If left uncorrected, liquid attenuation can lead
to errors in preciptiation rate estimates up to 3 mm hr$^{-1}$ in this example (Fig. 6f). Fig. 6 also
shows reflectivity and precipitation rate for the XSAPR2 which, as discussed in th previous
section, only suffer from negligible attenuation (Fig 6g and h). Comparing observations from
the unattenuated XSAPR2 (Fig. 6h) and observations from the KaSACR2 corrected for both
gas and liquid attenuation (Fig. 6d) also highlights the fact that even after all correction are
performed the KaSACR2 "realized" sensitivity does not allow it to detect some of the
precipitation the more sensitive XSAPR2 can detect. The range-dependent sensitivity of both
sensors can be contrasted in Fig. 9c.
**5.0 Complementary of different radar systems in Characterizing Light Precipitation**
**Variability**
As discussed in section 2.0, the KAZR2, KaSACR2 and XSAPR2 radars sample light
precipitation using very different transmission and sampling strategies. In this section we highlight
some of the advantages and tradeoffs of using each radar system to characterize different aspects
of light precipitation variability.
For illustration purposes, we compare, over the course of 36 hours between 00:00 UTC February
2 and 12:00 UTC February 3, hourly precipitation rate variability in the forms of frequency of
occurrence in different precipitation rate bins (pdf). Figure 8a shows estimates from the scanning
XSAPR2 collecting observation in PPI mode covering a domain between 2.5 and 40 km at 1°
elevation thus transecting heights between ~100 m and 750 m (also refer to Fig. 9a to visualize the
XSAPR2 sampling geometry). Figure 8b and c respectively show estimates from the vertically
pointing KAZR2 200 m above the surface and 90 m below cloud base which was around 850 m.
From Fig. 8b and c, it is evident that KAZR2, with its high sensitivity, is especially well suited to
document light precipitation and drizzle falling at a rate as low as 10$^{-4}$ mm hr$^{-1}$. KAZR2





observations show a reduction in the number of precipitation events and in precipitation intensity
from cloud base (Fig. 8c) towards the surface (Fig. 8b). This supports previous hypothesis that at
the ENA a large fraction of the light precipitation falls in the form of virga (Ahlgrimm and Forbes,
2014; Yang et al., 2018). Under these circumstances, where the character of precipitation changes
dramatically with height and its intensity is very low (below $10^{-3}$ mmhr$^{-1}$), scanning radar
observation at a fixed elevation may become inadequate to characterize surface precipitation over
a large domain owing to Earth curvature effects. Fig. 9a illustrates the height above the surface of
a 1° elevation scan with distance away from the radar; at a distance of 10-20 km the radar beam is
already 250 m above the surface while at a distance of 20-30 km this same radar beam is now 500
m from the surface. This non-uniformity of the radar beam height with distance makes scanning
cloud radar observations at one elevation angle more adequate to document the character of
vertically uniform precipitation. The rapid sampling rate of the KAZR2 also allows it to describe
the vertical structure of precipitation variability at a high temporal (scales as short as 2s).
On the other hand, one drawback of vertically pointing KAZR2 observations is that they are limited
to sampling only those precipitation events advected overhead. It is not uncommon to temporally
average vertically pointing observation to create a proxy for domain average statistics, however as
depicted in Fig. 5 it may be difficult to address the domain representativeness of one-hour of
vertically pointing precipitation rate estimates. It can also be challenging to interpret the mesoscale
organization of the precipitation field using vertically pointing observations alone; Scanning
systems such as the XSAPR2 can help fill this gap. Figure 5c and d show XSAPR2 1° elevation
PPI scans collected at 10:00 am and 8:00 am respectively which corresponds to the center time of
the KAZR2 time-height observations presented in Fig. 5a and b. XSAPR2 can observe the
structure and scales of popcorn precipitation and squall line precipitation over a domain of roughly
2,500 km$^2$. In its current configuration, the XSAPR2 system can be used to document the
horizontal structure and temporal variability of light-to-moderate precipitation on scales of ~5
minutes. Referring back to Fig. 8a hourly precipitation rate pdfs, it is evident that by covering a
larger domain XSAPR2 is able to observe a larger number of near surface sporadic precipitation
events such as that observed on Feb 03 around 0:00 and of isolated deep convective events
responsible for more intense precipitation ($R > 3$ mm hr$^{-1}$) such as that observed on Feb 03 around
8:00.

Now constrasting the two scanning radar XSAPR2 and KaSACR2. Although the Ka-band
SACR experiences less sea-clutter than the X-band SAPR, because of needs for cloud
sampling, it only currently performs one PPI scan at 0.5° every 15 min which limits its
temporal resolution. In addition, based on their technical specifications (Table 1), the XSAPR2
single pulse radar sensitivity is approximately 10 dB higher than that of the KaSACR2 (Fig. 9c
blue and black line respectively) and that is before considering that the Ka-band SACR also suffer
from significantly more attenuation from atmospheric gases (Fig. 9c green line) and liquid water
which even if corrected for still decrease it's "realized" sensitivity. For all these reasons, we
conclude that the XSAPR2 is more suitable for characterizing light precipitation variability
over large domains.
**6.0 Gridded Domain Precipitation Rate Estimation**

One way for scanning radars to overcome some of the limitation of their scanning strategy



is to develop horizontal, two-dimensional, gridded maps of the radar observables and other
quantities (i.e. precipitation rate) using measurements collected at different elevations angles (i.e.,
construct constant altitude plan position indicator (CAPPI) maps). Here, gridded XSAPR2
CAPPI's are constructed as follows: We perform the polar to Cartesian transformation for each
individual reflectivity measurement using a standard atmosphere radio propagation model which
considers the height of the beam above the Earth surface, and the distance between the radar and
the projection of the beam along the Earth surface (Doviak and Zrnic, 1993). Using these Cartesian
coordinates each PPI is mapped on a 100 m horizontal grid for which each grid point is populated
using a triangulation technique (i.e., the nearest three observations are linearly interpolated to
populate the grid cell). Then, every 100 m in the horizontal, a grid point at constant altitude is
populated by i) a measured value if falling on an elevation where observations were collected or
otherwise ii) a weighted average of the gridded data from the three closest PPI; The weight being
the inverse horizontal distance from the grid location. The aforementioned adaptive *Z-R*
relationships are then applied to the Cartesian grid reflectivity observations to produce
precipitation rate CAPPI such as the one illustrated in Fig. 9b. Figure 9 shows a Cartesian
coordinate constant altitude plan position indicator (CAPPI) map of precipitation rate constructed
around an altitude of 500 m using XSAPR2 observations collected between 1 and 5° elevation
(Fig. 9a, red color). This figure also illustrates how scanning radar sensitivity is range dependent
such that weak precipitation rates can only be detected close to the sensor (Fig. 9b light grey
colors). Producing an unbiased assessment requires the application of a uniform sensitivity
threshold over the entire domain observed by the scanning radar which creates a tradeoff between
documenting a large domain and documenting weak precipitation events. As quantified in Fig. 9c
at a distance of 40 km the XSAPR2 is only capable of detecting precipitation events of intensity
larger than $10^{-2.8}$ mm hr$^{-1}$ and any desire to document weaker precipitation rate events would
further limit domain size.

**7.0 Domain Average Precipitation Rate - When do Temporal and Horizontal Precipitation Variability Converge?**

The addition of the XSAPR2 at the ENA observatory offers new insights into precipitation
variability and organization over a domain of 40-60 km radius around the size. However, the
XSAPR2 data record is not as long as the KAZR data record which now spans 5 years at the ENA
even totaling up to 7.5 years if we consider the Cloud, Aerosols, and Precipitation in the Marine
Boundary Layer (CAP-MBL) campaign that took place at the site from April 2009 until January
2011 (Wood et al., 2005). Because of their longer data record, profiling radar observations have
the potential to inform us about decadal precipitation variability both temporal and structural.
However, with vertically pointing observations, it is near impossible to disentangle temporal
evolution from horizontal structure. Classical approaches rely on Taylor hypothesis of frozen
turbulence to convert elapsed time to horizontal dimension using the horizontal wind speed
responsible for advecting cloud and precipitation overhead. While widely used, little research has
been conducted to determine the validity and limitations of this assumption (see Oue et al. (2016)
for a discussion on cloud fraction). In this section we seek to determine how long does one need
to observe precipitation advected overhead to gather statistical precipitation information
equivalent to that of an 80 km wide domain.
Precipitation rate reported by XSAPR2 over a domain of 40 km radius around the site at 1°



elevation are used to evaluate the representativeness of KAZR2 observations collected 200 m
above the surface. To remove any bias caused by variations in minimum performance of both
sensors, a minimum precipitation rate threshold of $10^{-2.8}$ mm hr$^{-1}$ is applied to both sensors
reflecting the detectability of the XSAPR2 over the selected domain. Statistics for both sensors are
estimated using different set averaging time intervals (30 min, 1 h, 3 h, 12 h and 24 h) which allows
us to monitor the temporal variability of domain-average precipitation rate. For XSAPR2, using a
sliding window, we average all 5-min PPI observations collected during the chosen time interval.
For KAZR, we center the time window on the XSAPR2 estimates and average all 2-s observations
collected during the chosen time interval.
Focusing on features such as the width, the minimum, maximum and modes of the precipitation
rate statistical distribution; Results indicate that neither 30 min nor 1h averaging of KAZR
precipitation rate estimates can be used to replicate the precipitation rate statistics corresponding
to those of domain averaged over 30 min (Fig. 10 left column). Figure 10's 3$^{rd}$ and 4$^{th}$ columns/3$^{rd}$
and 4$^{th}$ rows, suggests that longer time averages (3h and 12 h) of KAZR2 observations capture the
most frequently occurring precipitation mode of domain-average precipitation rate on 3h and 12 h
timescales. Convergence between XSAPR2 and KAZR2 time-average precipitation rate estimates
is best (in terms of root mean square error (RMSE)) when considering the variability of domain-
average precipitation rate over timescales of 12 h (RMSE 13.4%); 12-h average domain-average
precipitation rate pdf from XSAPR2 and 12-h average precipitation rate pdf from KAZR are
similar in both magnitude and mode location.
Although these results are estimated with few observational cases (30 days), they clearly suggest
that XSAPR2 observations are necessary to characterize short-term (< 3 h) domain-average
precipitation rate characteristics. They also suggest that longer-term (12 h) domain-average
precipitation rate characteristics can be estimated by averaging either XSAPR2 or KAZR2
observations using time-windows of similar lengths.
**8.0 Summary and Conclusions**
The ARM ENA observatory is the first island-based climate research facility equipped with
collocated radars and lidars capable of sampling light oceanic precipitation. Here we presented the
characteristics and first light observations from three state-of-the-art 2$^{nd}$ generation radar systems:
the Ka-band Zenith radar (KAZR2), the Ka-band scanning ARM cloud radar (KaSACR2) and the
X-band scanning ARM precipitation radar (XSAPR2),
One of the initial concerns of operating scanning cloud and precipitation radars over the ocean is
the impact of sea-clutter, especially at low-elevation angles. Nearly one hundred hours of clear sky
observations were used to characterize the properties of sea-clutter in KaSACR2 and XSAPR2
observations. Analysis of clear and cloudy skies periods and intercomparison of the meteorological
and non-meteorological echoes of the KaSACR2 made it possible to design a relatively simple
filtering technique to isolate precipitation echoes in XSAPR2 observations. In short, a threshold
on normalized coherent power (< 0.3) and on average (5x5 window) cross-correlation (< 0.55),
can mitigate second-trip echoes and sea-clutter echoes. Everything considered, we find that
XSAPR2 observations collected at 1° elevation, albeit suffering from more clutter contamination
than KaSACR2, offer the best compromise between clutter contamination and proximity to the





surface.
Measurement calibration is also essential to quantitative precipitation rate retrieval. We applied
Kollias et al. (2019) technique to calibrate KAZR2 radar reflectivity measurements using Parsivel
disdrometer and CloudSat observations. Because they were found to match, the same offset is
applied to the KaSACR2 observations. To calibrate the XSAPR2 reflectivity measurements we
relied on a statistical comparison with GPM Ku-band radar observations collected around the ENA
site. The analysis indicated no noticeable offset; thus, no calibration offset was applied to the
XSAPR2. These techniques could be used in the future as a supplement to the ARM radar
engineering group efforts to characterize the ENA radars reflectivity measurements.
We capitalized on the availability of closely collected (in both time and physical distance) KAZR2,
ceilometer lidar and XSAPR2 measurement to estimate precipitation rate. Precipitation rates
retrieved using the O'Connor et al. (2005) radar-lidar technique have the advantage of being
estimated without assumptions on the drizzle drop size distribution shape and can accommodate
changes in aerosol loading, liquid water path and evaporation. Unfortunately, for a lack of scanning
lidar observations, we cannot apply this technique to scanning radar observations. Instead, we
showed how relating the retrieved precipitation rates in the column to radar reflectivity can be used
to estimate adaptive (in both time and height) parameters that related observed radar reflectivity
($Z$) to precipitation rate ($R$) in the form $Z = \alpha R^{\beta}$ which can be applied to retrieve precipitation
rate of the domain covered by scanning cloud radars. We report these adaptive parameters for the
period between 01/10/2018 and 04/01/2018 which includes the second phase of the ACE-ENA
campaign. These adaptive parameters were showed to capture changes in drop size distribution
with height as well as temporal changes in the cloud field.
Throughout this work, comparing precipitation rate statistics estimated by all three sensors
highlighted the following:
1) Because of strong signal attenuation by gases and liquid at Ka-band, X-band radars are
more suited for precipitation mapping especially over large domains.
2) When the character of precipitation varies rapidly with height for instance owing to an
active evaporation process, zenith-pointing radars are more suited for precipitation
characterization;
3) However, zenith-pointing observations collected over periods shorter than 12h should not
be considered representative of a domain especially one as large as 2,500 km$^2$ (i.e., ~40 km
radius half circle).
4) Estimates of domain precipitation rate variability on timescale of 12 hours can be captured
by averaging 12h of zenith-pointing radar observations collected at 200 m above the
surface.
5) Shorter term domain precipitation rate variability can only be capture by scanning
precipitation radars.
6) Scanning sensors are also better suited to document sporadic and horizontal homogeneous
precipitation including precipitation presenting mesoscale organization.
In a nutshell, the considerable differences in precipitation rate statistics estimated by the XSAPR2
and KAZR2 challenge our ability to objectively estimate precipitation rate statistics over a domain



for applications such as evaluation of high-resolution model output. Factors such as instrument
sensitivity, sampling temporal resolution, sampling height and domain size should always be
considered when comparing model output to observations for example through the use of forward
simulators.
**Authors contributions**
K. Lamer coordinated the project, performed the intercomparisons between the precipitation rates
produced by the three radars and produced the final manuscript draft. P. Kollias supervised Z. Zhu
and B. Puigdomènech Treserras as they respectively analyzed the KAZR2 and both the KaSACR2
and XSAPR2 observations; Analysis steps included performing data post-processing, calibration
and precipitation rate retrievals. B. Puigdomènech Treserras also produced the CAPPI part of this
work. B. Isom and N. Bharadwaj provided a wealth of information about the radar system
characteristics as well as guidance on radar data calibration. All coauthors have read the
manuscript draft and have contributed comments.

**Acknowledgments**
K. Lamer contributions were supported by subcontract 300324 of the Pennsylvania State
University with the Brookhaven National Laboratory in support to the U.S. Department of Energy
(DOE) ARM-Atmospheric Science Research (ASR) Radar Science group. B. Puigdomènech
Treserras contributions were supported through a subcontract with the Brookhaven National
Laboratory in support to the ARM-ASR Radar Science group Z. Zhu contributions were supported
by the U.S. DOE ASR ENA Site Science award. B. Isom and N. Bharadwaj contributions were
supported by Pacific North West National Laboratory. P. Kollias contributions were supported by
the U.S. DOE under Contract DE-SC0012704.

**Data availability**
All ARM data streams are available online at: http://www.archive.arm.gov/discovery/. All
GPM data streams are available online at  https://pmm.nasa.gov/data-access/downloads/gpm.



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



**Tables**
Table 1 Specification of ARM ENA zenith and scanning second generation radars

|  | KAZR2 | | KaSACR2 | | XSAPR2 | |
|---|---|---|---|---|---|---|
| Frequency (MHz) | 34860 | | 35290 | | 9500 | |
| Peak power (kW) | 2.2 | | 2.2 | | 300 | |
| Maximum Duty cycle (%) | 5.0 | | 5.0 | | 0.1 | |
| Pulse compression | Yes | | Yes (but not on) | | No | |
| Pulse length | 4 µs | 200 ns | ? | | 0.66 µs | |
| Sensitivity single pulse (dBZ) | -32.5 (at 1 km) | -44 (at 1 km) | -15 (at 20 km) | | -21 (at 20 km) | |
| Dead zone (m) | 72 | 737 | 400 | | 100 | |
| Unambiguous range (km) | 18 | | 40 | | Over 100 | |
| Gate spacing (m) | 30 | | 30 | | 100 | |
| Antenna size (m) | 1.82 | | 1.82 | | 5.0 | |
| 3-dB Beam width (°) | 0.3 | | 0.3 | | 0.45 | |
| Scan rate (° s$^{-1}$) | - | | 3 | | 6 | |
| Scan strategy | Zenith | | PPI scan | | VCP scan | |
| Elevation angle (°) | 90 | | 0.5 | | 0 to 5 every 0.5 | |
| Azimuthal sector (°) | - | | 360 | | 160 | |
| Scan time | 3 s | | 2 min | | 5 min | |
| Scan Interval | Continuous | | 15 min | | | |
| Transmit polarization | H | | Alternating H and V | | Simultaneous H and V | |
| Received polarization | H and V | | H and V | | H and V | |
| Amplifier Type | Klystron (EIKA) | | Klystron (EIKA) | | Magnetron | |
| Signal processing | FFT | | Pulse-pair | FFT | Pulse-pair | FFT |
| Doppler spectra | Yes | | No | Yes | No | Yes |
| Second trip echo removal technique | Challenging | | Frequency Hopping | Challenging | None | Coherent Power technique |
| Velocity dealiasing technique | Challenging | | Staggered Pulse Repetition Time | Challenging | Challenging | |






## Figures

**Figure 1.** For significant echoes, 1) radar reflectivity, 2) correlation coefficient ($\rho_{HV}$) and 3) relative frequency of occurrence of clutter as observed by the a) XSAPR2 at 0.5° elevation, b) XSAPR2 at 1° elevation and c) KaSACR at 0.5° elevation. d) Clutter characteristics estimated using 93 hours of clear sky observations.



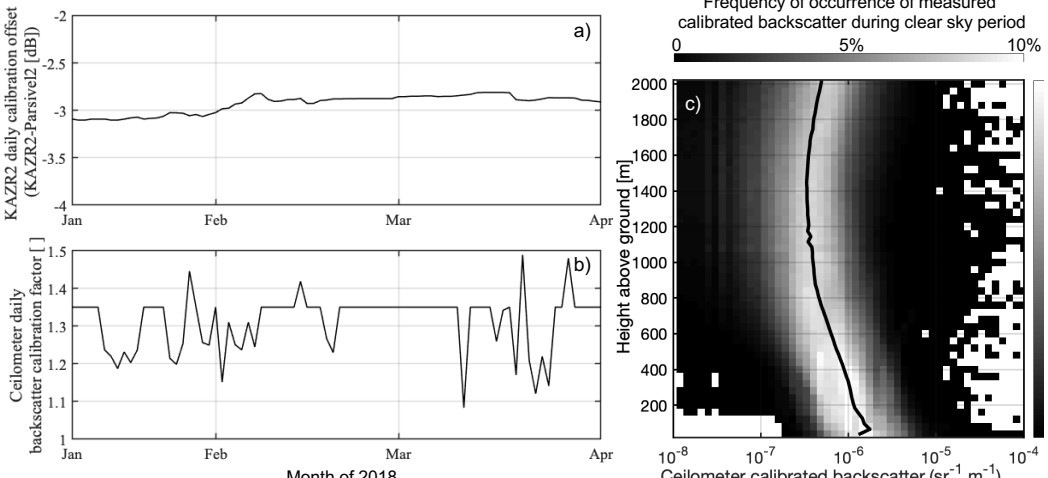

**Figure 2.** a) Ka-band Zenith Radar (KAZR) calibration offset to be removed from the KAZR radar reflectivity in order to match Parsivel Disdrometer radar reflectivity estimates. b) Ceilometer lidar calibration factor to be multiplied to observed backscatter to match theoretical liquid cloud lidar ratios. c) Frequency of occurrence of observed backscatter during clear sky conditions, solid black line is interpreted as the mean aerosol backscatter signal, observations small than this threshold at each height are eliminated from the drizzle analysis.

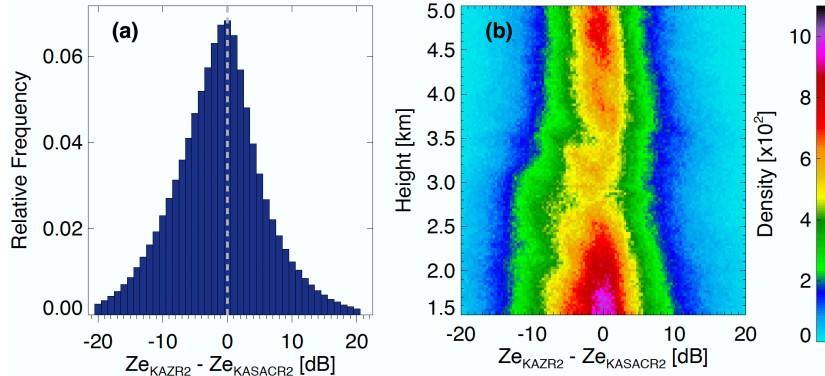

**Figure 3**. For period when KAZR2 and KaSACR2 are matched in time and range a) Difference in radar reflectivity reported by both sensors over the ranges between 1.5 and 5.0 km, b) Difference in radar reflectivity reported by both sensors as a function of range.



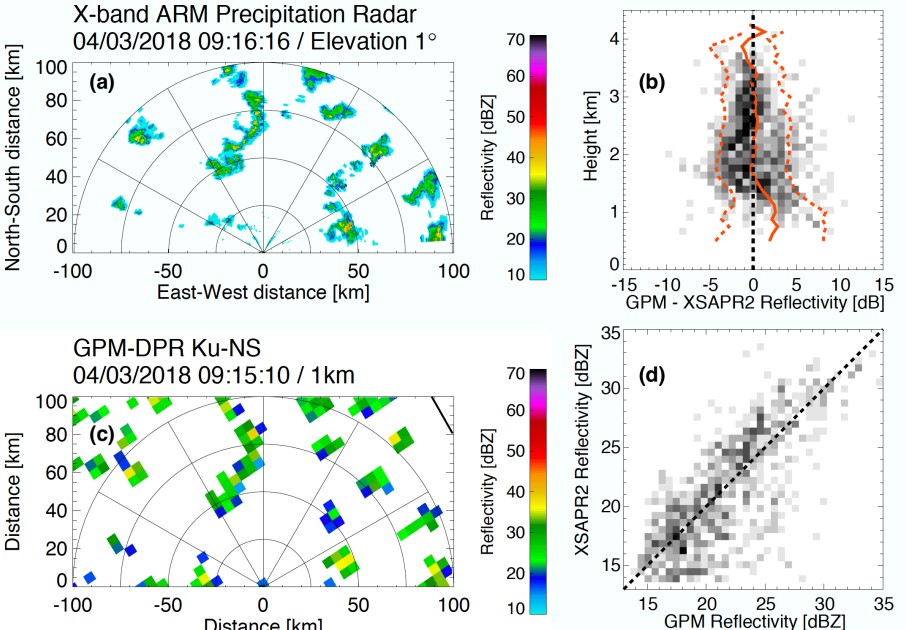

**Figure 4.** For the conditions that occurred on 04/03/2018 around 09:15 as observed by a) XSAPR2
radar reflectivity at 1° elevation and c) GPM-DPR Ku-band radar reflectivity at 1 km height. For
the entire geometry-matching dataset with 1516 points used for the calibration b) Difference
between the GPM-DPR Ku-band and XSAPR2 radar reflectivity measurements as a function of
height and d) scatterplot comparing the XSAPR2 and GPM-DPR Ku-band reflectivities
measurements.

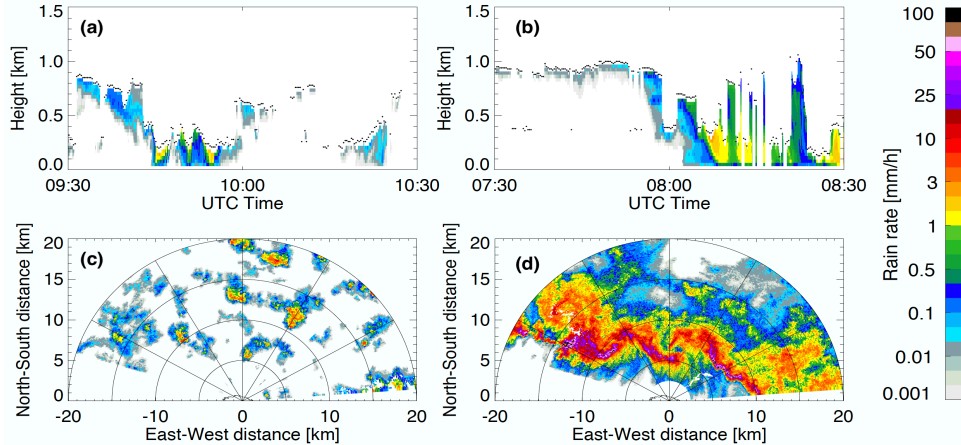

**Figure 5.** Retrieval of popcorn convection precipitation rate on 02/02/2018 using a) KAZR2
(zenith between 9:30 to 10:30 UTC) and c) KaSACR2 (1° elevation PPI at 10:00 UTC). Retrieval
of squall line precipitation rate on 03/02/2018 using b) KAZR2 (zenith between 7:30 to 8:30 UTC)
and d) KaSACR2 (1° elevation PPI at 8:00 UTC). Also indicated are the location of cloud bases
(black dots in panels a-b). Note that KAZR2 is located at (0 km,0 km).



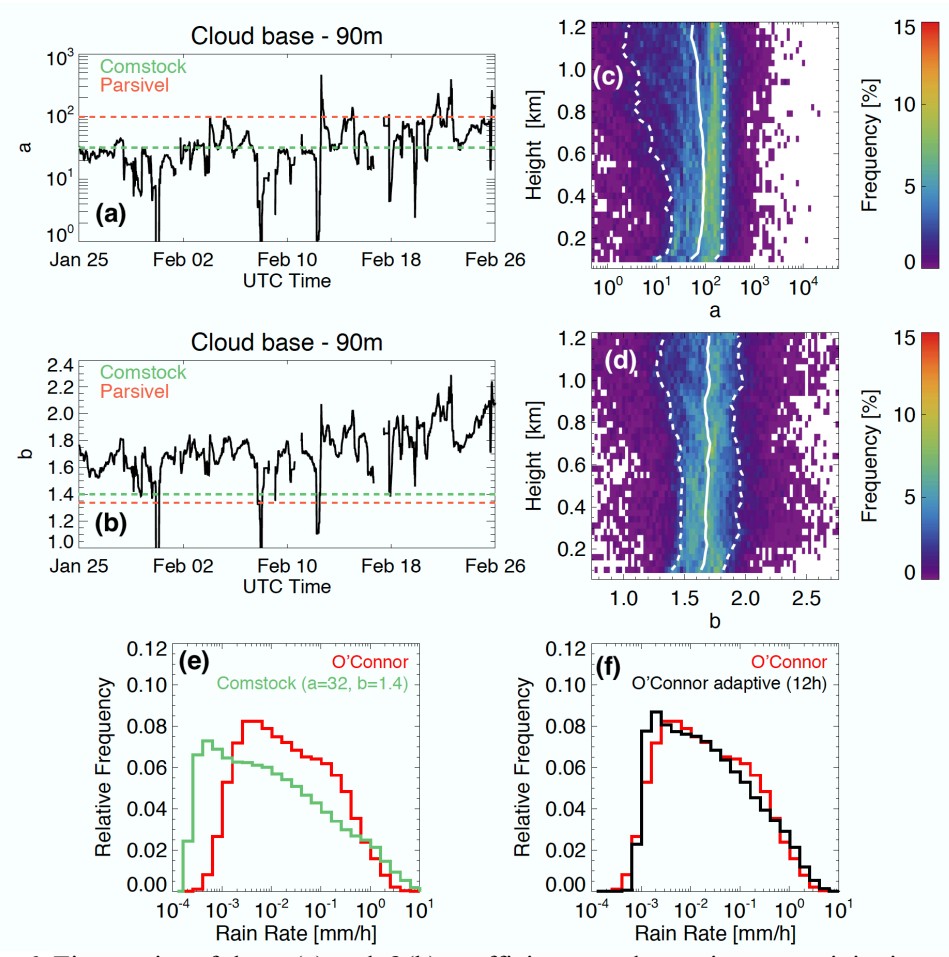

**Figure 6.** Time series of the $\alpha$ (a) and $\beta$ (b) coefficients used to estimate precipitation rate 90 m below cloud base height for a 30-day long period that overlaps with the second phase of the ACE-ENA field campaign. For the same time period, distribution of the $\alpha$ (c) and $\beta$ (d) coefficients with height along with their median (solid line) and 25th and 75th percentile values (dashed line). Precipitation rate distributions retrieved using the O'Connor et al. (2005) technique (red) and estimated using the adaptive coefficients (f, black) or the fixed coefficients proposed by *Comstock et al.*, [2004] (e, green). *Comstock et al.*, [2004] coefficients and coefficients determined from disdrometer observations are both presented in panels a and b using dashed green lines and orange lines respectively.

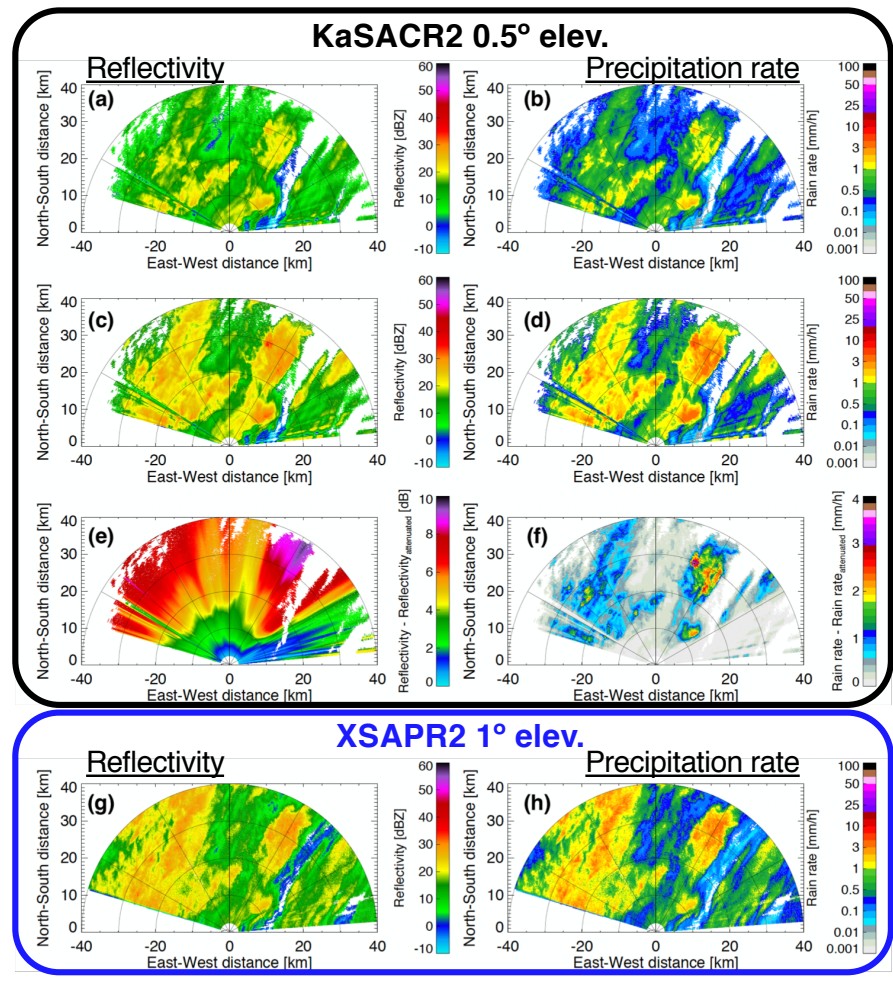

929
**Figure 7.** Example of observations/retrievals of the conditions happening on 02/13/2018 at
00:10 UTC. Shown for KaSACR2 perfoming 0.5° elevation PPI a) radar reflectivity field
corrected for gaseous attenuation neglecting liquid water attenuation and b) corresponding
precipitation rate retrieved using adaptive Z-R relationships; c) radar reflectivity field
corrected for both gas and liquid water attenuation and d) corresponding precipitation rate; e)
difference between a and c showing the range-accumulated radar reflectivity liquid water
attenuation correction and f) the corresponding precipitation rates bias. The upper panels (g)
and (f) show simultaneously collected XSAPR2 1.0° PPI observations for reference.
938
939

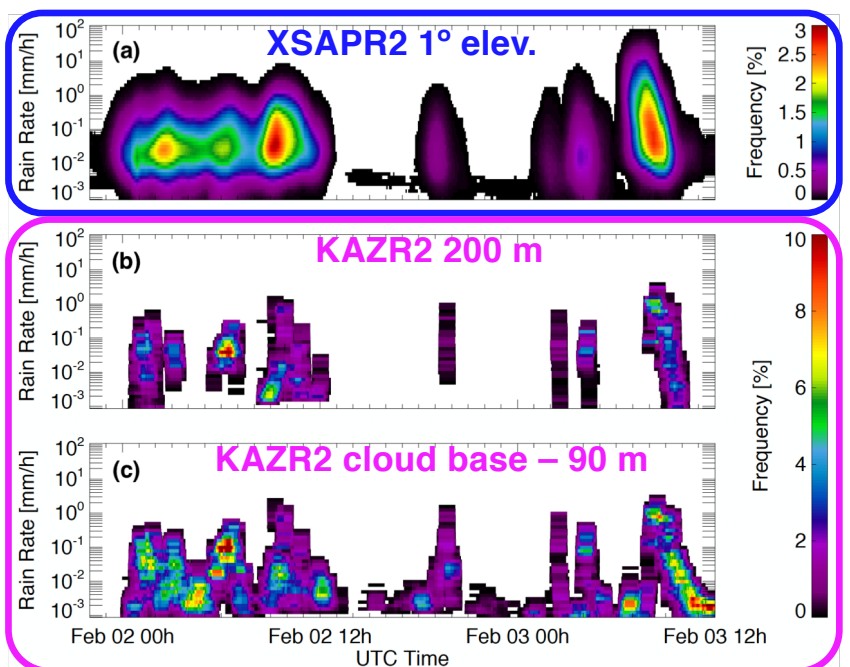

**Figure 8.** For a 36-h period (00:00 UTC February 2 to 12:00 UTC February 3), hourly probability density functions (pdfs) of precipitation rate estimated from a) XSARP2 when performing a 1 ° elevation PPI scan, b) KAZR2 200 m from the surface and c) KAZR2 90 m below cloud base height

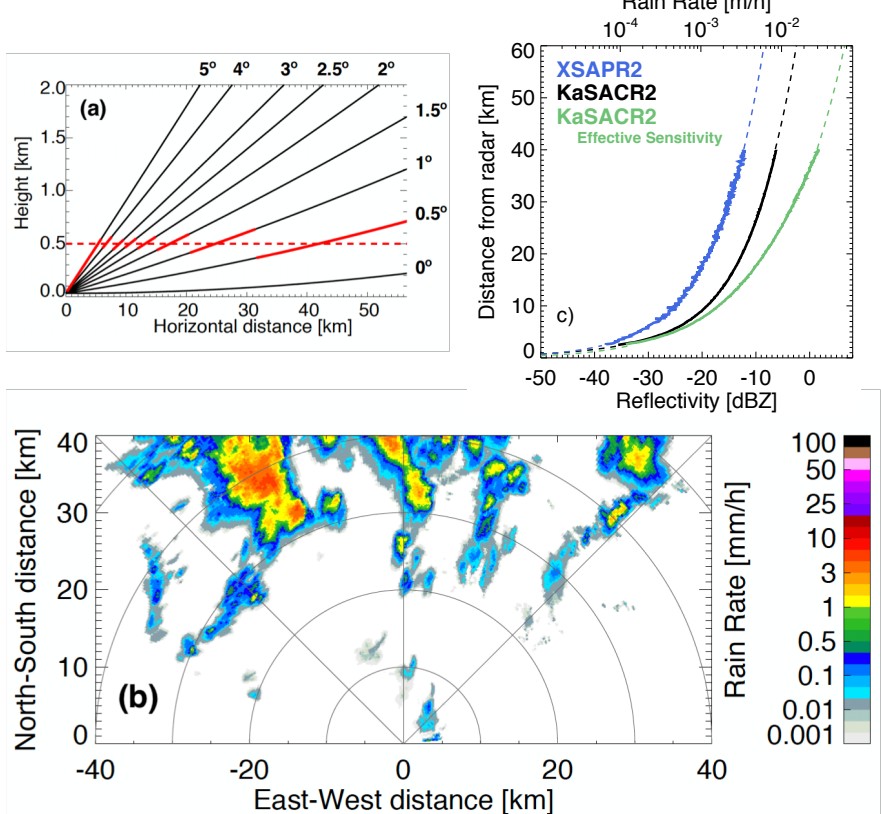

**Figure 9.** a) PPI scan geometry and b) Cartesian coordinate constant altitude plan position
indicator (CAPPI) map of precipitation rate constructed around an altitude of 500 m using
XSAPR2 observations collected 21/02/2018 on at 15:00 between 1 and 5° elevation. c) Theoretical
sensitivity of the XSAPR2 (blue) and KaSACR2 (black) along with the KaSACR2 "effective"
sensitivity considering it is affect by gas attenuation (green).

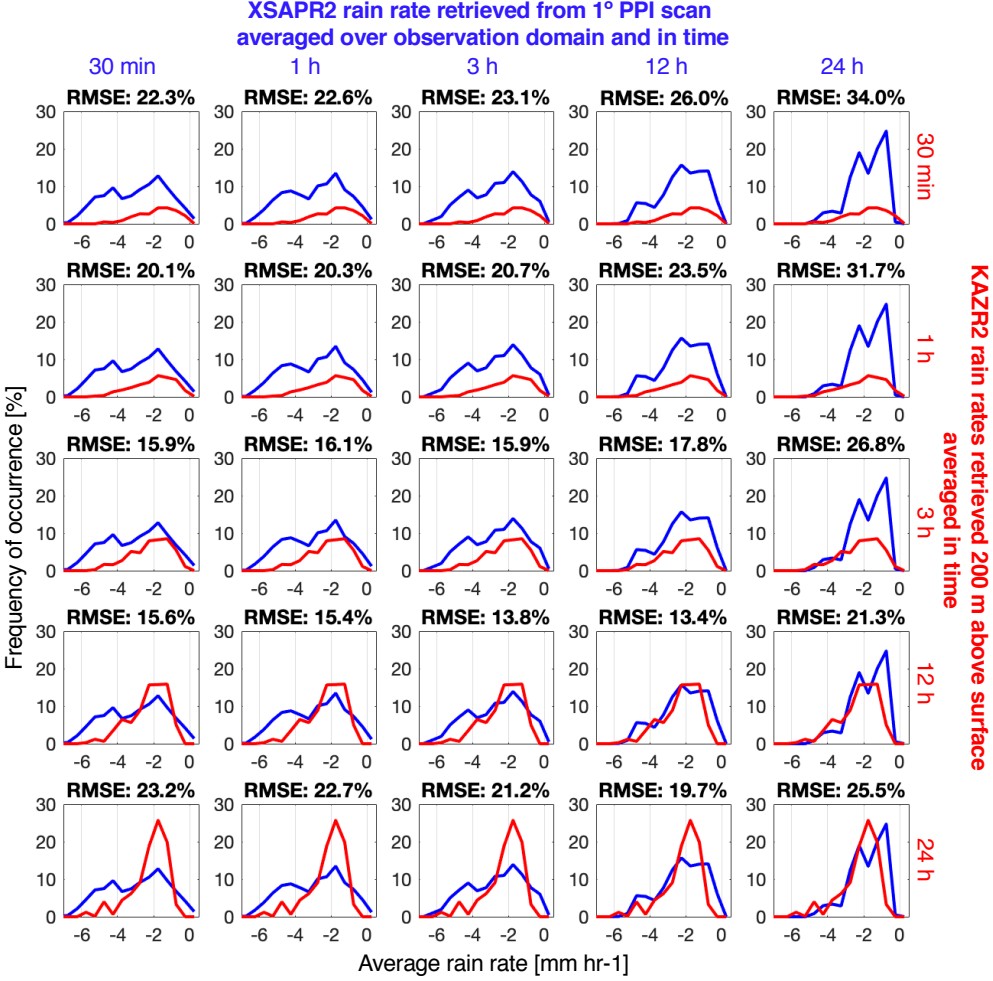

**Figure 10.** Probability density function of average (over different time windows) precipitation rate as estimated the XSAPR at 1° elevation over the domain between 2.5 and 40 km (blue) and as estimated by the KAZR2 at 200 m (red). Over each box is the root mean square error (RMSE) on the frequency of occurrence of precipitation rate estimated in 0.5 mm hr$^{-1}$ bins between -8 and 0.5 mm hr$^{-1}$.