# Peer review of "Characterization of Shallow Oceanic Precipitation using Profiling"

_Atmospheric Measurement Techniques, 2019_

## Referee Comment (RC1) · Anonymous Referee #1 · 2 Jun 2019

This paper develops a methodology for retrieving rainfall rates using data from two vertically pointing radars and a scanning radar at the ENA site in the Azores that is operated by the DOE ARM program. Overall, I think the rainfall rate retrieval using the adaptive Z-R relationship the and is a valuable contribution to AMT. This paper could be used as a citation for the ARM instrument mentors in the development of rainfall VAD products over the ENA. I do have a few concerns with the paper though before I would say that it is fit for publication in EMT.

One major concern is that in their calibration of the XSAPR2 data they state that there is no significant bias between the GPM reflectivities and the XSAPR2 reflectivities. However, in their own scatter plot, XSAPR2 looks to be about +2 or 3 dB hotter for reflectivities greater than about 25 dBZ, but it's hard to tell without applying statistical fits and tests. I am concerned that the agreement XSAPR2 and GPM at higher reflectivities (and hence higher rain rates) may not be as clear cut as is suggested in the paper.

My other major concern is that the authors mention that "considerable differences in precipitation rate statistics estimated by XSAPR2 and KAZR2 challenge our ability to objectively estimate precipitation statistics over a domain." I do not quite agree with this statement. The authors themselves have even established that XSAPR2 will provide better statistics simply due to the greater spatial coverage of XSAPR2. I think you can easily say that XSAPR2 is the better choice for deriving rainfall statistics simply due to its spatial coverage and reduced attenuation compared to KAZR2. So, I would like the authors to further clarify how these considerable differences between the two somehow complicate rainfall retrievals, because I honestly see a clear cut choice here.

The figures are also referred to out of order. For example, Figure 9 is referred to before Figure 6, which made it confusing for me to follow the figures. I would ask the authors in the next draft to place the figures in the order that they are referred to first in the paper. Also, there are incorrect references to Figure 7. I would urge the authors in the next draft to ensure that the Figures are also referred to correctly.

**Major comments:**
Line 37: Are you missing a "these" here? Right now you are suggesting that observations in general cannot produce objective estimates of precipitation, which is definitely not the case for every single situation.

Line 79: Is the lack of signal in KDP, ZDR simply a consequence of a narrower DSD that would be expected during the warm rain process?

Lines 316-319: I do not agree that there is no significant bias shown in this scatter plot. Figure 4d does look like there is a high bias in XSAPR2 when Z > 25 dBZ. Is it possible that the DPR data are contaminated by attenuation? Given the short wavelength I would think this would be a possibility. I think a more careful examination of this comparison is warranted.

Line 357-360: It actually looks like a lot of precipitation reaches the surface in Figure 5b, especially after 8 UTC. Could you please clarify in what conditions there is a more active evaporation process?

Section 4.3: Why were two different tilts of KaSACR2 and XSAPR2 used here? These two radars could be showing areas scanned that are 0.5 km apart. Also, there are several incorrect figures references in this section that need to be cleaned up.

Line 413: You mention that the two-way gas attenuation of XSAPR2 is negligible. However, attenuation from liquid at X-band can be significant, especially in the isolated deep convective cells. Have you applied any corrections for attenuation to the Z values in the development of your adaptive technique? Perhaps attenuation is not a major issue for the lighter precipitation events commonly observed at ENA, but I would foresee it being an issue in the isolated deep convective cases. Therefore, I think it's necessary to factor in the potential effects of liquid attenuation in your analysis.

Line 642-647: The considerable differences that we see are simply due to the very different samples that these instruments take. KAZR2 takes a soda straw view of the convection while XSAPR2 retrieves a full 3D volume. In addition, KAZR2 will be more heavily attenuated in heavy precipiation than XSAPR2. Therefore, these two p.d.f.s do not represent the same regions within the convection, and in general I would expect KAZR2 to not be as statistically representative of the observations simple due to the much lower sample volume you're factoring in. So it's not a surprise that the statistics are so different for lower averaging intervals. Have you tried to compare the statistics where the two are scanning the same spot? For example, by comparing the statistics over a single gate of XSAPR2 that is directly over KAZR2?

**Minor comments:**

Abstract line 34:
I would say the domain in terms of x by y km, not in km^2. This is generally more intuitive to the reader.

Line 59: Run-on sentence here.

Line 245: Extra "-" here.

Line 301: "XSAPR2."

Line 316: Though should be "although."

Figure 1: Your figures are not quite inside the boxes. Honestly, I would just remove the boxes around the figures.

---

## Referee Comment (RC2) · Anonymous Referee #2 · 27 Jun 2019

The paper by Lamer et al. entitled "Characterization of Shallow Oceanic Precipitation using Profiling and Scanning Radar Observations at the Eastern North Atlantic ARM Observatory" documents the precipitation retrievals from the 2nd generation radars at the ARM ENA site including KAZR2, KaSACR2 and XSAPR2. This paper describes the procedures for the radar post-processing procedures and provides some suggestions for different radars. And then it introduces a radar reflectivity-based precipitation retrieval technique using adaptive (in both time and height) parameters for scanning radars. After proposing that XSAPR2 is more suitable for characterizing the light precipitation over a large domain compared with KaSACR2, this paper demonstrates a gridded domain precipitation rate data reconstructed from XSAPR2 retrievals. Lastly,

the paper estimates the representativeness of zenith radar (KAZR2) observations and concludes that "the zenith radar observations cannot produce objective domain precipitation estimate and that forward-simulators should be used to guide high temporal-resolution model evaluation studies." The paper provides a valuable summary of radar observations and descriptive reference for ARM data users interested in the ENA precipitation retrievals, which has never been done before to my knowledge. However, the paper could benefit from some clarification and some parts need further development before publication in AMT. General comments: 1. I think the conclusion about the KaSACR2 precipitation rate would be more convincing if the paper shows some statistical analysis for a longer time period in addition to the theoretical sensitivity curve (Figure 9c) and one snapshot (Figure 7). Some further statistics would also help us better understand the bias of the KaSACR2 precipitation rate for marine boundary layer cloud regime. 2. It is not clear to me what time period, what weather conditions, and how many data samples are included in the analyses of Section 7. 3. This paper uses the XSAPR2 precipitation rate over a domain of 40 km radius around the site at 1° elevation and the KAZR2 precipitation rate at 200m above the surface to estimate the representativeness of zenith radar retrieved precipitation rate (Section 7). We know that the altitude of the XSAPR2 measurement increases with distance away from the radar (Figure 9a); and the XSAPR2 precipitation rate includes both horizontal and vertical variability (Figure 8), especially the vertical variability of the precipitation rate is pretty large in marine boundary layer cloud regime (e.g. Figure 5a). Therefore, this comparison is not just temporal vs. horizontal precipitation variability. I was not sure how to explain the convergence of these precipitation estimates at 12h and longer time scales shown in Figure 10. The paper demonstrates a gridded domain precipitation rate produce reconstructed from the XSAPR2 measurement in section 6 (Figure 9b). I wonder why this paper doesn't use the gridded data to estimate the representativeness of zenith radar retrieved precipitation rate. Also, I'd suggest the authors calculate the correlation coefficient between these two precipitation estimates, which provides more information about the relationship between these two precipitation estimates.

[Figure]

Specific comments: 1. I've noticed some typos scattered throughout the manuscript, so I'd recommend a close readthrough before resubmission. 2. Line 59: This sentence (and a few other sentences) should be separated into two sentences. 3. Line 95: "retrieved" –> "retrieve" 4. Line 101: "The ENA" –> "ENA" or "The ENA observatory" 5. Line 324-325: This sentence seems unnecessary to me. 6. Line 353: "In additional to" –> "In addition to" 7. Line 424: The referred figure jumps from Fig. 6 to Fig. 9. 8. Line 433-437: The Figure number is wrong (I guess it should be Figure 7). 9. Line 620: "were showed to" –> "were shown to" 10. Figure 4. The red lines in (b) have not been defined in the caption. 11. Figure 5. Can you add the main wind direction on (c) and (d)? It may help us better understand the results from the zenith radar and the scanning radar. 12. Figure 6(c). I'm not sure why the solid line (median) is away from the higher frequency of occurrence range (the orange color a = 1.5e2) between z = 0.8km and z = 1.2 km. 13. Figure 7. "The upper panel" –> "The bottom panel" 14. Figure 10. The x axis of the subpanels and the caption "precipitation rate estimated in 0.5 mm hr-1 bins between -8 and 0.5 mm hr-1": I don't understand why there are negative precipitation rates in the results. 15. The paper argues that "forward-simulators should be used to guide high temporal-resolution model evaluation studies" without providing any information about forward-simulators. I would suggest the authors to briefly describe what forward-simulators are and cite a few relevant references.

---

## Author Comment (AC1) · 19 Jul 2019

The authors would like to thank both reviewers for their insightful comments and for taking the time to report the many small typos which were unfortunately not caught by the author team. A point by point responds to the reviewer's comments can be found below.

**Reviewer 1**

One major concern is that in their calibration of the XSAPR2 data they state that there is no significant bias between the GPM reflectivities and the XSAPR2 reflectivities. However, in their own scatter plot, XSAPR2 looks to be about +2 or 3 dB hotter for reflectivities greater than about 25 dBZ, but it's hard to tell without applying statistical fits and tests. I am concerned that the agreement XSAPR2 and GPM at higher reflectivities (and hence higher rain rates) may not be as clear cut as is suggested in the paper.

Lines 316-319: I do not agree that there is no significant bias shown in this scatter plot. Figure 4d does look like there is a high bias in XSAPR2 when Z > 25 dBZ. Is it possible that the DPR data are contaminated by attenuation? Given the short wavelength I would think this would be a possibility. I think a more careful examination of this comparison is warranted

The concerns raised by the reviewer are valid and the authors agree that they should further discuss the caveats associated with such a cross-validation approach and slightly modify their conclusions. Please find below relevant excerpts from the revised manuscript. Note that we now do not refer to this procedure as a "calibration" procedure but rather to a "cross-validation methods". Moreover, it is worth nothing that most observations used in this comparison have reflectivity less than 25 dBZ. Data density is now displayed next to the revised c and d panels of Figure 4.

"Calibrating the XSAPR2 radar reflectivity measurements is more challenging since it does not perform profiling observations and as such it cannot be benchmarked against disdrometer and KAZR2 observations. Performing a physical subsystem calibration remains the best way to calibrate the XSAPR2 system. Prior to the ACE-ENA field campaign (06/2017) the ARM engineering team performed such a procedure which is expected to bring the calibration of the XSAPR2 system used in this study to within 1 dB. Here, in an effort to develop alternative calibration/cross-validation methods, we also compare the XSAPR2 radar observations to Global Precipitation Measurement (GPM) Ku-band frequency of the Dual-frequency Precipitation Radar (DPR) observation when the satellite track crosses within a 245 km radius of the XSAPR2 radar site. It is not expected that both sets of observations will perfectly match because of the different footprints, path lengths and surface returns of both radars but this comparison should at least provide some insight in the event that the difference between both sensors is larger than several dB. […]

Beyond agreeing in their location, both radars (XSAPR2 and GPM DPR) are found to agree on the reflectivity intensity of these precipitation echoes. To confirm their agreement, we estimated Contour of Frequency by Altitude Diagram (CFAD) of the differences in radar reflectivities between the matched XSAPR2 and GPM DPR for all 1516 available observations (Fig. 4b). Above the height at which GPM DPR is known to suffer from surface echo contamination (i.e., 1.5 km), the comparison between XSAPR2 reflectivities and GPM DPR reflectivities shows no

noticeable difference (i.e., no bias). A scatter plot between the matched GPM DPR and XSAPR2 radar reflectivity for height above 1.5 km confirms the overall lack of bias beyond the expect 1 dB between the two radars at all reflectivity (Fig. 4d on which the orange line depicts the best fit to the data and the dashed line represent a perfect match between the datasets and the grey shading indicates the data density). As mentioned above, scatter is expected because of the differences in configuration of both radar systems. The cloud types present in the cases available could further enhance the impact of the radar system differences since the shallow clouds observed during the 3 overpasses are of similar or even smaller size compared to the GPM DPR footprint. Small clouds could lead to non-uniform beam filling issue and as such to the GPM DPR underestimating the reflectivity of these cloud system which could partially explain the seemingly "high" bias of the XSAPR2 in Fig. 4d. Knowing that the ARM engineering team had calibrated the XSAPR2 just before the observations used here were collected and because this comparison with the GPM DPR showed no bias larger than several dB we conclude that, for the observation period between 01/10/2018 to 04/01/2018, the XSAPR2 was reasonably well calibrated and does not require any radar reflectivity adjustments."

[Figure]

**Figure 4.** For the conditions that occurred on 04/03/2018 around 09:15 as observed by a) XSAPR2 radar reflectivity at 1° elevation and c) GPM-DPR Ku-band radar reflectivity at 1 km height. For the entire geometry-matching dataset with 1516 points used for the calibration b) Scatter, mean (orange) and standard deviation (dashed lines) of the difference between the GPM-DPR Ku-band and XSAPR2 radar reflectivity measurements as a function of height and d) scatterplot comparing the XSAPR2 and GPM-DPR Ku-band reflectivities measurements above the GPM surface echo height of 1.5 km; Also plotted is the 1-to-1 relationship (dashed line) and the best linear fit to the observations (solid orange line).

My other major concern is that the authors mention that "considerable differences in precipitation rate statistics estimated by XSAPR2 and KAZR2 challenge our ability to objectively estimate precipitation statistics over a domain." I do not quite agree with this statement. The authors themselves have even established that XSAPR2 will provide better statistics simply due to the greater spatial coverage of XSAPR2. I think you can easily say that XSAPR2 is the better choice for deriving rainfall statistics simply due to its spatial coverage and reduced attenuation compared to KAZR2. So, I would like the authors to further clarify how these considerable differences between the two somehow complicate rainfall retrievals, because I honestly see a clear-cut choice here.

The authors agree with the reviewer that for the most part "Because of strong signal attenuation by gases and liquid at Ka-band, X-band radars are more suited for precipitation mapping especially over large domains.". However, we want to acknowledge the one caveat: "When the character of precipitation varies rapidly with height for instance owing to an active evaporation process, zenith-pointing radars are more suited for precipitation characterization".

The figures are also referred to out of order. For example, Figure 9 is referred to before Figure 6, which made it confusing for me to follow the figures. I would ask the authors in the next draft to place the figures in the order that they are referred to first in the paper. Also, there are incorrect references to Figure 7. I would urge the authors in the next draft to ensure that the Figures are also referred to correctly.

We apologize to the reviewer for the mix-up in figure references. Figures are now referred to in order and are properly referred to in the text.

**Major comments:**

Line 37: Are you missing a "these" here? Right now you are suggesting that observations in general cannot produce objective estimates of precipitation, which is definitely not the case for every single situation.

The reviewer is correct, the word "these" was added. Thank you.

Line 79: Is the lack of signal in KDP, ZDR simply a consequence of a narrower DSD that would be expected during the warm rain process?

The reviewer is correct, additional clarification is now given in the manuscript: "Beyond detecting, quantifying the spectrum from drizzle to rain from warm clouds is especially challenging since at small drizzle rates the droplets they contain are mostly spherical and as such do not generate the typical polarimetric signals required of common precipitation rate retrievals (e.g., Villarini and Krajewski, 2010; Gorgucci et al., 2000)."

Line 357-360: It actually looks like a lot of precipitation reaches the surface in Figure 5b, especially after 8 UTC. Could you please clarify in what conditions there is a more active evaporation process?

The reviews question is very interesting however we believe that documenting the conditions that lead to more or less drizzle evaporation is somewhat beyond the scope of this study which is focused on describing updated radar systems and on describing a precipitation retrieval technique.

Looking in literature we would say that our Figure 5a shows conditions consistent with Yang et al. (2018) study of single-layer marine stratocumulus clouds conducted in the Eastern North Atlantic where they report that drizzle is a common feature of marine stratocumulus cloud and that most of the drizzle drops evaporate in the subcloud layer before reaching the ground. In their study based on 42 days of stratocumulus cloud observations collected over a year, they found that 83% of the cloud profiles were drizzling with only 31% of them generating precipitation reaching the surface.

On the other hand, our Figure 5b shows a different scenario with a squall line probably associated with a cloud field deeper than a stratocumulus deck. The more intense rain produced by such cloud systems is most likely to not completely evaporate before reaching the surface; However, the gradient from green to blue seen in Figure 5b does support our statement that "the most intense precipitation rates are observed near cloud base height".

Section 4.3: Why were two different tilts of KaSACR2 and XSAPR2 used here? These two radars could be showing areas scanned that are 0.5 km apart.

Although we agree with the reviewer that it would be optimum to compare KaSACR2 and XSAPR2 observations collected at the same 0.5° elevations tilt, our analysis of the prevalence of clutter in the XSAPR2 0.5° elevations tilt (Section 3.1) lead us to conclude that "Given this, XSAPR2 cross validation and precipitation rate maps will be estimated using observations collected at 1.0° elevation since it offers the best compromise between proximity to the surface and minimum sea-clutter contamination." Unfortunately, KaSACR2 solely collected observations at 0.5° elevations tilt thus not allowing for a comparison between XSAPR2 and KaSACR2 1.0° elevations tilt.

We revised the text to reflect this reality:

"With the caveat that we are comparing rain rates retrieved at slightly different slanted elevations, comparing rain rates retrieved from the XSAPR2 observations (Fig. 8h) and from the KaSACR2 observations corrected for both gas and liquid attenuation (Fig. 8d) also highlights the fact that even after all correction are performed the KaSACR2 "realized" sensitivity does not allow it to detect some of the precipitation the more sensitive XSAPR2 can detect."

Line 413: You mention that the two-way gas attenuation of XSAPR2 is negligible. However, attenuation from liquid at X-band can be significant, especially in the isolated deep convective cells. Have you applied any corrections for attenuation to the Z values in the development of your adaptive technique? Perhaps attenuation is not a major issue for the lighter precipitation events commonly observed at ENA, but I would foresee it being an issue in the isolated deep

convective cases. Therefore, I think it's necessary to factor in the potential effects of liquid attenuation in your analysis.

We completely agree with the reviewer that the decision of applying or not a liquid attenuation correction highly depends on the type of precipitation system. Text was added to the revised manuscript to reflect this comment.

"Note how the adaptive *Z-R* relationships were directly applied to clutter-filtered calibrated XSAPR2 radar reflectivity measurements since we estimate that, for the majority of the conditions occurring at the ENA observatory, both two-way gas attenuation and liquid attenuation at X-band are negligible; According to Rosenkranz (1998), at X-band frequency, gas attenuation generally amounts to 0.03 dB km$^{-1}$ which is much smaller than even the radar calibration uncertainty. Similarly, Matrosov et al. (2005) discusses how, for rain rates of 2 mm hr$^{-1}$, liquid attenuation roughly amounts to 0.015 dB km$^{-1}$ which over the depth of the shallow systems producing this type of precipitation cumulates to liquid attenuation less than 1 dB again within the radar calibration uncertainty. We do however acknowledge that, for deep convective systems, liquid attenuation correction would be granted, but since this type of precipitating system was not being frequently observed at the ENA observatory, we did not apply any liquid attenuation correction to the XSAPR2 measurements."

Line 642-647: The considerable differences that we see are simply due to the very different samples that these instruments take. KAZR2 takes a soda straw view of the convection while XSAPR2 retrieves a full 3D volume. In addition, KAZR2 will be more heavily attenuated in heavy precipitation than XSAPR2. Therefore, these two p.d.f.s do not represent the same regions within the convection, and in general I would expect KAZR2 to not be as statistically representative of the observations simple due to the much lower sample volume you're factoring in. So, it's not a surprise that the statistics are so different for lower averaging intervals. Have you tried to compare the statistics where the two are scanning the same spot? For example, by comparing the statistics over a single gate of XSAPR2 that is directly over KAZR2?

Our intent is not to match the XSAPR2 and KAZR2 volume rather it is to confront this reality:

"The addition of the XSAPR2 at the ENA observatory offers new insights into precipitation variability and organization over a domain of 40-60 km radius around the size. However, the XSAPR2 data record is not as long as the KAZR data record which now spans 5 years at the ENA even totaling up to 7.5 years if we consider the Cloud, Aerosols, and Precipitation in the Marine Boundary Layer (CAP-MBL) campaign that took place at the site from April 2009 until January 2011 (Wood et al., 2005). Because of their longer data record, profiling radar observations have the potential to inform us about decadal precipitation variability both temporal and structural. However, with vertically pointing observations, it is near impossible to disentangle temporal evolution from horizontal structure. Classical approaches rely on Taylor hypothesis of frozen turbulence to convert elapsed time to horizontal dimension using the horizontal wind speed responsible for advecting cloud and precipitation overhead. While widely used, little research has been conducted to determine the validity and limitations of this assumption (see Oue et al. (2016) for a discussion on cloud fraction). In this section we seek to determine how long does one need to observe precipitation advected overhead to gather

statistical precipitation information equivalent to that of an 80 km wide domain."

We attempt to address the difference in sensitivity of both system by "a minimum precipitation rate threshold of $10^{-2.8}$ mm hr$^{-1}$ is applied to both sensors reflecting the detectability of the XSAPR2 over the selected domain.". Moreover, we limit the comparison domain to "40 km radius around the site".

Following the reviewers comment, to improve at least the vertical collocation of both systems, we revised our approach as follows: "Although any height could be used, we perform this comparison at the specific height of 500 m; While KAZR2 precipitation retrievals can be directly extracted at 500 m, those from XSAPR2 must be extracted from gridded CAPPI fields which are constructed following the details provided in Section 6 using a collection of PPI scans."

This new approached yield very similar conclusions:

"Focusing on features such as the width, the minimum, maximum and modes of the precipitation rate statistical distribution; Results indicate that neither 30 min nor 1h averaging of KAZR precipitation rate estimates can be used to replicate the precipitation rate statistics corresponding to those of domain averaged over 30 min (Fig. 10 left column). Averaging of 3 hrs of KAZR2 data improves its representativeness of domain average rain rate variabilities on scales of 1 to 3-hrs (2nd and 3rd rows/3rd column). Convergence between XSAPR2 and KAZR2 precipitation rate estimates is simingly best when considering the variability of domain-average precipitation rate over 12 h (correlation coefficient R=0.25) or longer timescales; 12-h average domain-average precipitation rate pdf from XSAPR2 and 12-h average precipitation rate pdf from KAZR are similar in both magnitude and mode location."

"When it comes to capturing the general shape of the precipitation rate distribution, 12-hrs of zenith-pointing radar observations can be averaged to represent the 12-h variability of such a ~40 km radius half circle domain ."

[Figure]

**Figure 10.** Probability density function of average (over different time windows) precipitation rate as estimated the XSAPR and by the KAZR2 (red) both at 500 m above the surface in $10^{0.5}$ mm hr$^{-1}$ bins; The XSAPR2 precipitation rates 500 m above the surface being from gridded CAPPI constructed using a collection of PPI scans and are limited to the domain between 2.5 and 40 km around the location of the KAZR2. Over each box is the correlation coefficient (R) between the XSAPR2 and the KAZR2 average precipitation rates.

**Minor comments:**

Abstract line 34: I would say the domain in terms of x by y km, not in km^2. This is generally more intuitive to the reader.

We agree with the reviewer; we now refer to the domain as a "40-km radius half circle"

Line 59: Run-on sentence here.

The sentence was broken down in two and slightly shortened: "Quantification, over a domain of

several kilometers, of marine drizzle cell precipitation rate and environmental conditions, could provide additional observational constrains for modeling studies. Unfortunately collecting such observations remain challenging over the ocean."

Line 245: Extra "-" here.

We would like to thank the reviewer for reporting to typo. It was corrected.

Line 301: "XSAPR2."

We would like to thank the reviewer for reporting to typo. It was corrected.

Line 316: Though should be "although."

We would like to thank the reviewer for reporting to typo. It was corrected.

Figure 1: Your figures are not quite inside the boxes. Honestly, I would just remove the boxes around the figures.

Figure 1 was reproduced without the boxes.

---

## Author Comment (AC2) · 19 Jul 2019

The authors would like to thank both reviewers for their insightful comments and for taking the time to report the many small typos which were unfortunately not caught by the author team. A point by point responds to the reviewer's comments can be found below.

**Reviewer 2**

General comments:

1. I think the conclusion about the KaSACR2 precipitation rate would be more convincing if the paper shows some statistical analysis for a longer time period in addition to the theoretical sensitivity curve (Figure 9c) and one snapshot (Figure 7). Some further statistics would also help us better understand the bias of the KaSACR2 precipitation rate for marine boundary layer cloud regime.

We agree with the reviewer that a larger dataset would help further determine the potential of the KaSACR2 for precipitation characterization. However, here where both KaSACR2 and XSAPR2 observations were collected, we want to make the point that, simply from the standpoint of the radar specification, the XSAPR2 system is much more suited for precipitation studies:

"Now constrasting the two scanning radar XSAPR2 and KaSACR2. Although the Ka-band SACR2 experiences less sea-clutter than the X-band SAPR2, because of needs for cloud sampling, it only currently performs one PPI scan at 0.5° every 15 min which limits its temporal resolution. In addition, based on their technical specifications (Table 1), the XSAPR2 single pulse radar sensitivity is approximately 10 dB higher than that of the KaSACR2 (Fig. 9c blue and black line respectively). Finally, the Ka-band SACR2 also suffer from significantly more attenuation from atmospheric gases (Fig. 9c green line) and liquid water which even if corrected for still decrease it's "realized" sensitivity. For all these reasons, we conclude that the XSAPR2 is more suitable for characterizing light precipitation variability over large domains."

We made sure to revise our final conclusions specifying that those apply to the XSAPR2 and not the KaSACR2:

" 5) Shorter term domain precipitation rate variability can only be capture by scanning precipitation radars and especially those operating at weakly-attenuating frequencies and with high sensitivity such as the XSAPR2

6) Scanning sensors such as the XSAPR2 are also better suited to document sporadic and horizontal homogeneous precipitation including precipitation presenting mesoscale organization."

2. It is not clear to me what time period, what weather conditions, and how many data samples are included in the analyses of Section 7.

We agree with the reviewer that it would be appropriate the restate the size of our dataset as it is relevant to the analysis of Section 7. We now specified in Section 7: "Over the 3-month period between 01/10/2018 and 04/01/2018, the domain representativeness of KAZR2 precipitation rate

estimates is evaluated using XSAPR2 observations collected over a domain of 40 km radius around the site.".

3. This paper uses the XSAPR2 precipitation rate over a domain of 40 km radius around the site at 1∘ elevation and the KAZR2 precipitation rate at 200m above the surface to estimate the representativeness of zenith radar retrieved precipitation rate (Section 7). We know that the altitude of the XSAPR2 measurement increases with distance away from the radar (Figure 9a); and the XSAPR2 precipitation rate includes both horizontal and vertical variability (Figure 8), especially the vertical variability of the precipitation rate is pretty large in marine boundary layer cloud regime (e.g. Figure 5a). Therefore, this comparison is not just temporal vs. horizontal precipitation variability. I was not sure how to explain the convergence of these precipitation estimates at 12h and longer time scales shown in Figure 10. The paper demonstrates a gridded domain precipitation rate produce reconstructed from the XSAPR2 measurement in section 6 (Figure 9b). I wonder why this paper doesn't use the gridded data to estimate the representativeness of zenith radar retrieved precipitation rate. Also, I'd suggest the authors calculate the correlation coefficient between these two precipitation estimates, which provides more information about the relationship between these two precipitation estimates.

The reviewer's comment is a very good one. To this effect, we recomputed the pdfs using XSAPR2 500 m CAPPI precipitation rates and KAZR2 500 m precipitation rates and now report the correlation coefficient between the two. While the results differ somewhat, our conclusions change very little. Please find below our revisions of this section of the manuscript:

"Over the 3-month period between 01/10/2018 and 04/01/2018, the domain representativeness of KAZR2 precipitation rate estimates is evaluated using XSAPR2 observations collected over a domain of 40 km radius around the site. Although any height could be used, we perform this comparison at the specific height of 500 m; While KAZR2 precipitation retrievals can be directly extracted at 500 m, those from XSAPR2 must be extracted from gridded CAPPI fields which are constructed following the details provided in Section 6 using a collection of PPI scans. […]

Focusing on features such as the width, the minimum, maximum and modes of the precipitation rate statistical distribution; Results indicate that neither 30 min nor 1h averaging of KAZR precipitation rate estimates can be used to replicate the precipitation rate statistics corresponding to those of domain averaged over 30 min (Fig. 10 left column). Averaging of 3 hrs of KAZR2 data improves its representativeness of domain average rain rate variabilities on scales of 1 to 3-hrs (2nd and 3rd rows/3rd column). Convergence between XSAPR2 and KAZR2 precipitation rate estimates is seemingly best when considering the variability of domain-average precipitation rate over 12 h (correlation coefficient R=0.25) or longer timescales; 12-h average domain-average precipitation rate pdf from XSAPR2 and 12-h average precipitation rate pdf from KAZR are similar in both magnitude and mode location.

Although these results are estimated with few observational cases (3-month period), they clearly suggest that XSAPR2 observations are necessary to characterize short-term (< 1 h) domain-average precipitation rate characteristics. They also suggest that longer-term (12 h) domain-average precipitation rate characteristics can be estimated by averaging either XSAPR2 or KAZR2 observations using time-windows of similar lengths."

"When it comes to capturing the general shape of the precipitation rate distribution, 12-hrs of zenith-pointing radar observations can be averaged to represent the 12-h variability of such a ~40 km radius half circle domain ."

[Figure]

**Figure 10.** Probability density function of average (over different time windows) precipitation rate as estimated the XSAPR and by the KAZR2 (red) both at 500 m above the surface in $10^{0.5}$ mm $hr^{-1}$ bins; The XSAPR2 precipitation rates 500 m above the surface being from gridded CAPPI constructed using a collection of PPI scans and are limited to the domain between 2.5 and 40 km around the location of the KAZR2. Over each box is the correlation coefficient (R) between the XSAPR2 and the KAZR2 average precipitation rates.

Specific comments:

1. I've noticed some typos scattered throughout the manuscript, so I'd recommend a close readthrough before resubmission.

We would like to apologize to the reviewer for our oversights. We were more careful as we revised the manuscript.

2. Line 59: This sentence (and a few other sentences) should be separated into two sentences.

The sentence was broken down in two and slightly shortened: "Quantification, over a domain of several kilometers, of marine drizzle cell precipitation rate and environmental conditions, could provide additional observational constrains for modeling studies. Unfortunately collecting such observations remain challenging over the ocean."

3. Line 95: "retrieved" –> "retrieve"

We would like to thank the reviewer for reporting to typo. It was corrected.

4. Line 101: "The ENA" –> "ENA" or "The ENA observatory"

Changed for "The Eastern North Atlantic region".

5. Line 324-325: This sentence seems unnecessary to me.

The sentence was removed.

6. Line 353: "In additional to" –> "In addition to"

We would like to thank the reviewer for reporting to typo. It was corrected.

7. Line 424: The referred figure jumps from Fig. 6 to Fig. 9.

We apologize to the reviewer for the mix-up in figure references. Figures are now referred to in order and are properly referred to in the text.

8. Line 433-437: The Figure number is wrong (I guess it should be Figure 7).

We apologize to the reviewer for the mix-up in figure references. Figures are now referred to in order and are properly referred to in the text.

9. Line 620: "were showed to" –> "were shown to"

We would like to thank the reviewer for reporting to typo. It was corrected.

10. Figure 4. The red lines in (b) have not been defined in the caption.

We apologize for the oversight. The red lines in b depict the mean and standard deviation. The figure caption was revised accordingly.

11. Figure 5. Can you add the main wind direction on (c) and (d)? It may help us better understand the results from the zenith radar and the scanning radar.

The general wind direction was added using arrows in panels c-d.

12. Figure 6(c). I'm not sure why the solid line (median) is away from the higher frequency of occurrence range (the orange color a = 1.5e2) between z = 0.8km and z = 1.2 km.

We verified and the position of the solid white line reflect the mean of the measurements at each height, the median is to the left of the region with the highest density of points since the distribution is skewed.

13. Figure 7. "The upper panel" –> "The bottom panel"

We would like to thank the reviewer for reporting to typo. It was corrected.

14. Figure 10. The x axis of the subpanels and the caption "precipitation rate estimated in 0.5 mm hr-1 bins between -8 and 0.5 mm hr-1": I don't understand why there are negative precipitation rates in the results.

Rain rate are reported in logarithmic scale and the caption should read "precipitation rate estimated in $10^{0.5}$ mm hr$^{-1}$ bins between $10^{-8}$ and $10^{0.5}$ mm hr$^{-1}$." We would like to apologize to the reviewer for the confusion.

15. The paper argues that "forward-simulators should be used to guide high temporal-resolution model evaluation studies" without providing any information about forward-simulators. I would suggest the authors to briefly describe what forward-simulators are and cite a few relevant references.

We agree with the reviewer that additional information is granted, we added to following material to the revised manuscript:

"Factors such as instrument sensitivity, sampling resolution, sampling height and domain size should always be considered when comparing model output to observations. One way to consider these factors could be to convert model output rain rates to observable rain rate through the use of forward simulators which can use drop size and atmospheric conditions information to reproduce the attenuation affecting radar signals. Several forward-simulator further take into consideration the dependency of radar sensitivity with range which dictates the minimum detectable rain rate at various distance within a domain (e.g., Tatarevic et al., 2015; Lamer et al., 2018)."